# Winter drainage of surface lakes on the Greenland Ice Sheet from Sentinel-1 SAR Imagery

Corinne Benedek[1] and Ian Willis[1]

[1]University of Cambridge, CB2 1ER, UK

**Correspondence:** C.L. Benedek (clb90@cam.ac.uk)

**Abstract.** Surface lakes on the Greenland Ice Sheet play a key role in its surface mass balance, hydrology, and biogeochemistry. They often drain rapidly in the summer via hydrofracture, which delivers lake water to the ice sheet base over timescales of hours to days and then can allow melt water to reach the base for the rest of the summer. Rapid lake drainage, therefore, influences subglacial drainage evolution, water pressures, ice flow, biogeochemical activity, and ultimately the delivery of water, sediments and nutrients to the ocean. It has generally been assumed that rapid lake drainage events are confined to the summer, as this is typically when observations are made using satellite optical imagery. Here we develop a method to quantify backscatter changes in satellite radar imagery, which we use to document the drainage of six different lakes during three winters (2014/15, 2015/16 and 2016/17) in fast flowing parts of the Greenland Ice Sheet. Analysis of optical imagery from before and after the three winters supports the radar-based evidence for winter lake drainage events and also provides estimates of lake drainage volumes, which range between $0.000046 \pm 0.000017$ km$^3$ and $0.0200 \pm 0.002817$ km$^3$. For three of the events, optical imagery allows repeat photoclinometry (shape from shading) calculations to be made showing mean vertical collapse of the lake surfaces ranging between $1.21 \pm 1.61$ m and $7.25 \pm 1.61$ m, and drainage volumes of $0.002 \pm 0.002968$ km$^3$ to $0.044 \pm 0.009858$ km$^3$. For one of these three, time-stamped ArcticDEM strips allow for DEM differencing, which demonstrates a mean collapse depth of $2.17 \pm 0.28$ m across the lake area. The findings show that lake drainage can occur in the winter in the absence of active surface melt and notable ice flow acceleration, which may have important implications for subglacial hydrology and biogeochemical processes.

## 1 Introduction

Lakes form each summer on the surface of the Greenland Ice Sheet (GrIS), particularly in the upper ablation and lower accumulation areas (McMillan et al., 2007; Selmes et al., 2011; Liang et al., 2012; Pope et al., 2016; Williamson et al., 2017). They enhance melt rates by reducing albedo (Lüthje et al., 2006; Tedesco et al., 2012), store water and delay its delivery to the ocean (Banwell et al., 2012; Leeson et al., 2012; Arnold et al., 2014), and collect nutrients - the products of surface inorganic and organic chemical processes (Musilova et al., 2017; Lamarche-Gagnon et al., 2019). Many lakes drain over the summer (Selmes et al., 2013; Williamson et al., 2017), sometimes slowly by overtopping their basins and incising a channel (Hoffman et al., 2011; Tedesco et al., 2013; Koziol et al., 2017) but often rapidly by hydrofracturing from the surface to the base of the ice sheet (Das et al., 2008; Doyle et al., 2013; Tedesco et al., 2013; Stevens et al., 2015; Chudley et al., 2019). The rapid

drainage of a lake may trigger the opening of crevasses and the generation of moulins (Hoffman et al., 2018) or the drainage of other lakes (Christoffersen et al., 2018) through ice dynamic coupling. Rapid lake drainage provides a major shock to the ice sheet as millions of cubic metres of water are delivered to the bed in a few hours, and the resultant fracture may permit meltwater to reach the bed for the rest of the summer. This lake drainage and subsequent water input generates a radiating subglacial water 'blister' beneath the draining lake, which evolves into a conduit in the down-hydraulic-potential direction allowing the lake water and subsequent melt water to be evacuated (Pimentel and Flowers, 2010; Tsai and Rice, 2010; Dow et al., 2015). High water pressures are generated transiently during lake drainage (Banwell et al., 2016), lifting the ice sheet off the bed and increasing temporarily its sliding velocity (Das et al., 2008; Doyle et al., 2013; Tedesco et al., 2013; Stevens et al., 2015; Chudley et al., 2019). The subsequent evolution of the subglacial conduit may lower water pressures (Schoof, 2010; Hewitt, 2013; Werder et al., 2013; Banwell et al., 2016) and reduce sliding speeds, often below pre-drainage values as a result of temporary increases in basal hydraulic efficiency (Bartholomew et al., 2010).

Rapid lake drainage and subsequent meltwater influx also alter subglacial biogeochemistry as large volumes of oxygenated water containing surface microbial taxa and inorganic and organic nutrients replace wintertime anoxic waters and associated microbes, shifting subglacial redox potential and associated biogeochemical pathways (Wadham et al., 2010; Shade et al., 2012). Thus, lake drainage events influence the quantity and quality of water issuing from the ice sheet, although their effects are superimposed on the larger scale atmospheric controls on melt patterns and runoff. They can produce small floods that flush out sediments (Bartholomew et al., 2011), raise levels of phosphorus, nitrogen and sulphate in proglacial streams (Hawkings et al., 2016; Wadham et al., 2016), and mark a transition from net subglacial methane production and proglacial export during winter to consumption with little or no export in the summer (Dieser et al., 2014).

Much of what we know about the locations, timings and magnitudes of rapid lake drainage events comes from the analysis of optical satellite imagery (Box and Ski, 2007; McMillan et al., 2007; Sneed and Hamilton, 2007; Leeson et al., 2013; Moussavi et al., 2016; Pope et al., 2016; Williamson et al., 2018) although studies have recently begun using optical imagery from drones (Chudley et al., 2019), and airborne and satellite radar data (Miles et al., 2017; Schröder et al., 2020). Conventional understanding is that rapid lake drainages are confined to the summer and may be driven by active in-situ hydrofracture through the lake bottom triggered by increased lake volume (Alley et al., 2005; van der Veen, 2007; Krawczynski et al., 2009; Arnold et al., 2014; Clason et al., 2015) and/or by passive fracture in response to perturbations in ice sheet flow induced by surface meltwater initially tapping the bed via nearby moulins (Stevens et al., 2015; Chudley et al., 2019). In this understanding, lakes completely or partially drain during the summer then freeze during the winter, either freezing through completely or maintaining a liquid water core (Selmes et al., 2013; Koenig et al., 2015; Miles et al., 2017; Law et al., 2020). High proglacial stream discharge anomalies outside of the summer melt season have been attributed to the release of stored water from the ice sheet (Rennermalm et al., 2013; Lampkin et al., 2020). On another occasion, proglacial stream evidence together with the appearance of surface collapse features on the ice sheet were used to suggest that water may have been released from surface lakes in January and February of 1990 (Russell, 1993). A recent study using satellite radar data has identified a few winter lake drainage events (Schröder et al., 2020).

Here we develop an algorithm to examine spatial and temporal variations in microwave backscatter from Sentinel-1 satellite synthetic aperture radar (SAR) imagery and document the location and timing of six separate lake drainage events over three different winters. We confirm the winter lake drainages and provide estimates of draining lake volumes through calculation of water areas and depths in Landsat-8 optical imagery from the previous and subsequent melt seasons. For three of the events, the optical imagery allows us to calculate surface elevation changes associated with the lake drainages using the technique of

photoclinometry. For one of those three events an independent calculation of surface elevation change is available through the comparison of time-stamped ArcticDEM strips before and after the event.

## 2   Methods

The study was conducted over a 30,452 km$^2$ area of the GrIS (Figure 1). The site spans elevations from 300 m to 2038 m above sea level and includes approximately 300 lakes over 5 pixels in size (0.0045 km$^2$). The study period spans imagery from

July 2014 to May 2017 and includes, therefore, three fall-winter-spring periods from October through May, hereafter "winter periods": 2014/15, 2015/16 and 2016/17.

There are six components to our analysis. First, a lake mask is established from optical imagery. Second, for each lake, trends in mean backscatter change during the winter are calculated. Third, the backscatter changes are used to identify large anomalous, sudden and sustained increases in backscatter that are indicative of winter lake drainage events. Fourth, optical

images from before the winter periods are used to provide estimates of lake volumes prior to drainage. Fifth, for three of the events, optical imagery and the technique of photoclinometry are used to calculate patterns of surface elevation change associated with the lake drainage events, providing independent estimates of lake drainage volumes. Sixth, for one of those three events, time-stamped ArcticDEM differencing is used to confirm the patterns of elevation change and provide another independent measure of lake drainage volume. These components to our analysis are described more fully in the six sections

below.

### 2.1   Establishing lake outlines using optical imagery

Prior to each winter, lake boundaries were delineated based on a calculation of maximum NDWI$_{\text{ice}}$ per pixel from optical imagery during the preceding late melt season (late July through August, image IDs listed in Appendix E). Landsat-8 Tier 1 TOA images were chosen based on minimal cloudiness (filtered using the Landsat-8 QA band) and images were removed from

the set manually where cloudiness interfered with NDWI$_{\text{ice}}$ calculations. Late season images were chosen so that lakes that had already drained prior to the end of summer freeze-over period were not included in the calculations. For each late summer period, multiple images were needed to cover the entire region and to obtain at least one cloud-free pre-freeze-over image for all areas of the study site.

Normalized Difference Water Index NDWI$_{\text{ice}}$ was calculated for each pixel in each of the images in the Landsat-8 set (Yang

and Smith, 2012) (Equation 1).

$$NDWI_{\text{ice}} = (Blue-Red)/(Blue+Red) \tag{1}$$

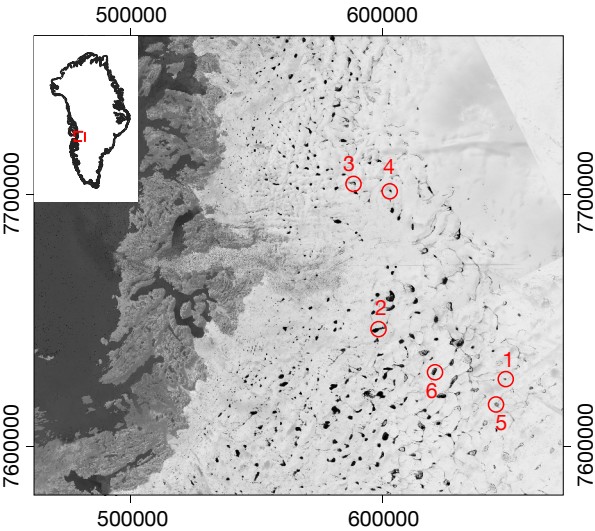

**Figure 1.** Study area within the context of the Greenland Ice Sheet (inset). Distribution of all surface lakes detected from optical imagery, with the six winter draining lakes highlighted (red numbers, in chronological order of drainage), which are shown in more detail in Figure 6. The base map is a composite image showing the maximum NDWI$_{ice}$ observed for each pixel in Landsat-8 optical images over the course of all summers from 2014 through 2017. The outline of Greenland is from OpenStreetMap (© OpenStreetMap contributors 2019. Distributed under a Creative Commons BY-SA License.)

where Blue and Red refer to band reflectance.

For each late summer, a mask was created from the set of Landsat-8 images by recording the maximum NDWI$_{ice}$ value observed in each pixel over the set and setting an NDWI$_{ice}$ threshold of 0.25 following Yang and Smith (2012) and Miles et al.
(2017) indicating the presence of deep water. These lake masks, one for each summer, were then used as the basis for defining lake boundaries for the analysis of backscatter changes in SAR imagery during the subsequent winter periods.

### 2.2   Calculating time series of mean lake backscatter from SAR imagery

For each winter period, lake masks delineated from the previous late summer's Landsat-8 images were applied to Sentinel-1 SAR images in order to calculate trends in mean backscatter for each lake over time. Analysis was restricted to lakes identified
in the optical data, as the delineation of lakes from SAR imagery alone is not trivial. Low backscatter values in C-Band SAR could be indicative of surface characteristics other than the expression of water. Changes in mean backscatter of each lake were tracked over each winter period and these changes were used to identify wintertime lake drainages as described further below.

Google Earth Engine (Gorelick et al., 2017) was used to select a series of Sentinel-1 images over the study site. Sentinel-1 images on the Google Earth Engine repository have been pre-processed using the following steps: i) Apply Orbit File; ii)
Thermal Noise Removal; iii) Radiometric Calibration (to Gamma Nought); iv) Terrain Correction (orthorectification using SRTM, to UTM 22 projection). We restricted our selection to ascending relative orbits to reduce backscatter variation from

image to image due to look angle alone. While Sentinel-1 has a repeat pass time of 12 days per satellite (6 days when both 1A and 1B satellites are combined), not all images are collected, sometimes leaving lengthy data gaps over the study site. For the purposes of this study, images from ascending Relative Orbit 17 were used as this orbit provided the greatest number of images over the study site within the study period. Three images were removed as outliers as they exhibited significant scene-wide departures from the backscatter of images adjacent in time. Both HH and HV polarizations are available for our study site, but we include only the data from the HV polarization as it more clearly shows buried shallow near-surface lakes (Miles et al., 2017). The presence of water may be observed even when the lake surface is frozen and covered by snow as the HV polarisation of C-band SAR can penetrate up to a few metres of ice (Rignot et al., 2001).

## 2.3   Isolating drainage events

To examine changes in lake behaviour, we created a time series of mean backscatter for each lake through each winter using Sentinel-1 imagery. Lakes undergo a slow freeze-through process over the winter (Selmes et al., 2013; Law et al., 2020). Water in C-Band SAR imagery presents as low backscatter. As the lake surface begins to freeze, scattering due to bubbles trapped in the ice increases. C-Band waves continue to reach the underlying water until the ice becomes thick enough to obscure it. Summer lake drainage events have been observed to follow a pattern of low to high backscatter (Johansson and Brown, 2012; Miles et al., 2017). A winter lake drainage would result in the same trend of low to high backscatter due to the removal of water and the exposure of the ice underneath, in addition to roughness added above by the collapse of the ice lid. We hypothesize, therefore, that a winter lake drainage event would appear as a large sudden increase in backscatter between two images, which is then sustained over a long period of time, in much the same way as it does for a summer lake drainage (Miles et al., 2017; Dunmire et al., 2020).

To be certain that a large sudden increase in mean backscatter is an expression of a change in a particular lake, rather than an artifact of the sensing process, an anomalous increase in lake backscatter is identified by comparing the mean backscatter change of each lake to that for all the other lakes in the scene in the same consecutive image pair. For a selection of lakes, the backscatter frequency distributions were examined and shown to be close to normally distributed and thus lake medians and means were close in value. For each consecutive image pair, the z-score of backscatter change for each lake is calculated relative to the backscatter change of all lakes within the study site and a threshold of +1.5 is used to isolate those lakes that experience a greater than average increase in backscatter between images.

To be sure that a large anomalous and sudden increase in backscatter was sustained rather than just an isolated occurrence, filters were employed to check for reversal in the subsequent three images, where those images occurred within 48 days of the last of the original pair. In each timestep, lakes were removed from consideration if the reversed backscatter change was greater than 25% of the magnitude of the original anomalous increase (see 'A' in Figure 2). Time series were also checked for a dip in backscatter prior to the large rise (see 'C' in Figure 2). In the instances where the magnitude of the dip was greater than 25% of the magnitude of the sudden increase, that lake was removed from consideration as a draining lake. The aim of this processing was to identify lakes that showed a sustained backscatter step change increase between two relatively stable levels (see also 'B' and 'D' in Figure 2). Given that there are some large gaps in Sentinel-1 data collection within each relative orbit,

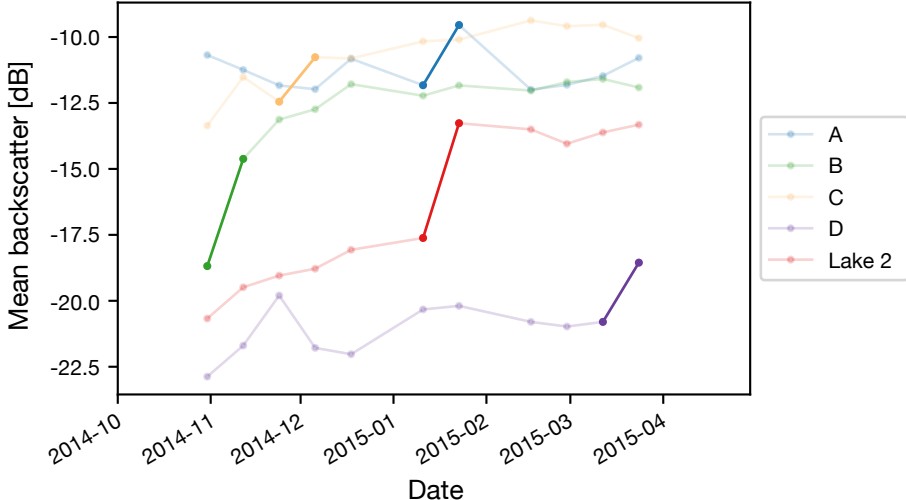

**Figure 2.** This figure illustrates the filtering criteria for identifying drained lakes. (A) Anomalous sustained step change but one that is not sustained. (B) Anomalous increase but with insufficient history to determine if the change was an adjustment from a previous dip or step increase from a previous low. (C) Anomalous sustained change but with a prior dip such that this change was a return to prior values rather than a sustained change. (D) Anomalous change without sufficient information to confirm a sustained change. Lake 2 shows anomalous sudden and sustained backscatter change depicting lake drainage. All the time series shown are results from actual lakes in the 2014-2015 season. Bold line segments are the transitions that met the z-score threshold.

specifying that a change event had to occur within 12 days and be sustained for up to 48 days, reduced the number of events compared to those originally detected. Finally, only lakes greater than 5 pixels in size (8000 m$^2$) were considered.

## 2.4 Lake volume

Lake depths were calculated from Landsat 8 imagery using physical principles based on the Bouguer-Lambert-Beer law as outlined elsewhere (Sneed and Hamilton, 2007; Pope et al., 2016; Williamson et al., 2018). For the six lakes we found that drained in the winter, the latest Landsat-8 images showing the lake prior to freezing over were selected manually. Lake depth, z, was calculated on a per-pixel basis from:

$$z = [ln(Ad - Rinf) - ln(Rpix - Rinf)]/g \tag{2}$$

where $A_d$ is the lake bottom albedo, Rinf is the reflectance of a deep water pixel, Rpix is the reflectance of the pixel being assessed, and g is based on calibrated values for Landsat 8 (Pope et al., 2016). For this analysis, calculations were performed for both the red and panchromatic bands with the final depths taken as the mean of the two results (Pope et al., 2016; Williamson et al., 2018). For each band, the outline of each lake was established using a mask based on an NDWI$_{ice}$ threshold of 0.25. The reflectance values of all pixels immediately exterior (30m) to this outline were averaged to obtain a value for $A_d$. Rinf

was determined per image by selecting the darkest pixel (which was always a seawater pixel). For each lake, the depths of all lake pixels were summed to calculate lake volume. Error in the depth calculation follows from Pope et al. (2016). We take the average of the documented error for the Landsat-8 red band (0.28 m) and that for the panchromatic band (0.63 m) to give an error of 0.46 m. Uncertainty in lake volume follows from this uncertainty in the depth calculation. In line with previous work, we do not define errors for lake areas, which instead are fixed according to our threshold $NDWI_{ice}$ value of 0.25.

## 2.5 Elevation change from photoclinometry

This technique is also known as 'shape-from-shading' and uses a single surface DEM and a Landsat-8 image to develop a relationship between reflectance and slope in a baseline location to then extrapolate the topography in another. We used photoclinometry to reconstruct the topography of the lake surface using winter Landsat-8 images before and after the drainage event and then produced a differencing image.

The ArcticDEM (5m resolution mosaic) (Porter et al., 2018) served as the base DEM for area surrounding the lake, and was resampled using bilinear interpolation to match the 30 m Landsat-8 resolution. Landsat-8 image pairs were chosen to be as close to the timing of each lake drainage as possible both before and after, as well as cloud free over the lake, and from the same Path and Row to reduce any incidence angle error. All images used were taken when the surface was snow covered to ensure that reflectance variation was due to surface slope. The calculations follow the methods outlined by Pope et al. (2013) and were completed for three of the six drained lakes as suitable Landsat-8 image pairs did not exist for the other three.

For each Landsat-8 image (six in total, two per lake) the following procedure was adopted. Band 4 was extracted and used as the basis for calculation. Transects were drawn across the lake parallel with the solar azimuth at the time of the image. Transects were 10 km in length, to achieve sufficient coverage of both the lake and ambient area, and were spaced 250 m apart across the width of the lake. The lake was outlined manually based on the Band 4 image, and a 100 m buffer external to the lake boundary was added to ensure that the changing lake topography was not included in the production of a baseline relationship between topography and reflectance. Each transect was sampled every 30 m along its length for Band 4 reflectance and for elevation in the ArcticDEM. Sample lake imagery is shown in Appendix A. Surface slope was calculated between each pair of sample points outside the buffer region along each transect. A linear relationship was established between slope and Band 4 reflectance for all sampled points outside the buffered lake area.

For each image processed, the linear slope-reflectance relationship established for non-lake pixels was then applied to the buffered lake pixels to calculate slope for each of the nodes on each transect across the buffered lake area. Elevation for each node on each transect across the buffered lake was reconstructed by integrating the slope values, starting from the known elevation of the node at the edge of the buffered lake on the north side of the lake and progressing to the south side. This resulted in small offset errors on each transect at the nodes on the south side of the buffered lake, where elevations did not match the known elevations from the DEM. These offsets were closed by linearly tilting each transect across the buffered lake, adjusting all elevations accordingly (Appendix A4). Elevation values were then interpolated (IDW method) using a 250 m x 30 m grid to create a digital elevation model of each lake before and after drainage. These grids were then differenced to calculate the patterns of lake surface elevation change due to winter lake drainage.

Error in the photoclinometry depth calculation is derived from Pope et al. (2013), who compared elevations derived using the photoclinometry method applied to Landsat imagery with airborne LiDAR elevation data. In areas where the photoclinometry assumptions were met (no shading) the median error was just 0.03 m, so the height difference error is 0.04 m. In areas where the photoclinometry assumptions were not always met (e.g. shaded areas), the median error was 1.44, so the height difference error is 1.61 m. We suspect the real error for our case on the Greenland Ice Sheet lies somewhere between these two, but to account for the different locations, DEMs, solar elevations and along-track spacings of the sample points between the Iceland and Greenland studies, we use the larger of the two errors, i.e. 1.61 m. As for the attenuation-based depth calculations, we do not define errors for lake areas, which are fixed according to our threshold $NDWI_{ice}$ value of 0.25.

## 2.6 ArcticDEM Differencing

We used 2 m time-stamped ArcticDEM strips (Porter et al., 2018) from dates prior to and after each drainage but within the winter season to avoid changes due to surface melt. Relevant DEMs could only be found for Lake 6 dated 21 Sept 2016 and 12 March 2017. We calculated the difference between these two DEMs in the region of Lake 6 to determine changes in surface elevation over this time period and an independent measure of drained lake volume.

Error in the ArcticDEM depth differential follows from Noh and Howat (2015). Error in the calculation of the DEM is approximately 0.2 m so the height difference error is 0.28 m.

## 3 Results

### 3.1 Winter lake drainage from Sentinel-1 imagery

We found six lakes that experienced large, anomalous, sudden and sustained backscatter increases that we interpret as lake drainage events over the three winter seasons analyzed. Three of these events (Lakes 2, 5 and 6) appear clear in the Sentinel-1 imagery and are supported by optical imagery and photoclinometry evidence with one of them (Lake 6) also supported by ArcticDEM differencing. The remaining three lakes exhibit a time series of mean backscatter change that is in line with our expectations of drained lake behaviour but have insufficient evidence from other datasets to confirm drainage.

The locations of the drained lakes are shown in Figure 1 and the drainage characteristics are summarized in Table 1. Although one of the criteria for lake selection was having a z-score of backscatter increase greater than 1.5, results show that all six lakes that met all of the criteria had a z-score of backscatter increase greater than 2.0 (Table 1). The size of the drained lakes varied widely (between 0.18 km$^2$ and 6.84 km$^2$) as did the timing of drainage within the winter season, ranging between early November and late February (Table 1). During the 2015-2016 winter, Lakes 3 and 4 towards the north of the study area, and separated by a straight-line distance of 14.9 km, drained within the same 12 day time period (Figure 1 and Table 1).

For each lake, the backscatter changes that signify a drainage are shown in Figure 3. All lakes generally undergo a large, anomalous, sudden change from predominantly dark (low backscatter) to light (higher backscatter) when compared to their surroundings. This transition is visually more obvious for the larger lakes (Lakes 1, 2, 5, and 6) and less clear for the smaller

**Table 1.** Details of the lake drainage events. Location refers to longitude, latitude (WGS84). The drainage dates are the Sentinel-1 image dates over which the anomalous change was identified. The delta dB is the mean change in backscatter (measured in decibels) within the lake boundary from one image to the next. The z-score is the measure of the magnitude of this backscatter change compared to the backscatter change of other lakes in the study site across the same image pair. Lake area is the size of the lake delineated by the $NDWI_{ice}$-based mask. Lake volume was calculated as described in Methods.

| Lake | Location | Drainage Date | delta dB | z-score | Pre-drainage Lake Area | Pre-drainage Mean Lake Depth | Pre-drainage Lake Volume |
|------|----------|---------------|----------|---------|------------------------|------------------------------|--------------------------|
| Lake 1 | -47.32 , 68.70 | 11 Nov 2014 to 23 Nov 2014 | -4.3 | 3.5 | $0.04\ km^2$ | $0.57 \pm 0.46\ m$ | $21,212 \pm 17\ m^3$ |
| Lake 2 | -48.52, 68.91 | 10 Jan 2015 to 22 Jan 2015 | -4.4 | 3.4 | $6.12\ km^2$ | $3.26 \pm 0.46\ m$ | $19,964,800 \pm 2817\ m^3$ |
| Lake 3 | -48.75, 69.43 | 05 Jan 2016 to 17 Jan 2016 | -3.8 | 2.7 | $0.43\ km^2$ | $1.89 \pm 0.46\ m$ | $809,000 \pm 197\ m^3$ |
| Lake 4 | -48.38, 69.40 | 05 Jan 2016 to 17 Jan 2016 | -2.3 | 2.6 | $0.51\ km^2$ | $2.56 \pm 0.46\ m$ | $1,318,400 \pm 237\ m^3$ |
| Lake 5 | -47.43, 68.62 | 10 Feb 2016 to 22 Feb 2016 | -3.2 | 2.8 | $1.84\ km^2$ | $0.86 \pm 0.46\ m$ | $1,593,600 \pm 848\ m^3$ |
| Lake 6 | -48.03, 68.75 | 06 Nov 2016 to 18 Nov 2016 | -9.3 | 2.2 | $2.27\ km^2$ | $1.41 \pm 0.46\ m$ | $3,188,800 \pm 1043\ m^3$ |

lakes (Lakes 3 and 4) (Figure 3) although the mean backscatter change for Lake 3 is actually slightly greater than that for Lake 5 (Table 1).

The mean backscatter time series for each lake is shown in Figure 4. Each series shows at least two dates of similar backscatter values prior to the step change from low to high backscatter. Each series maintains its higher backscatter after the initial jump. The backscatter changes of Lakes 3 and 4 are smaller in dB than the change that occurs in Lake 6 but the z-scores signifying how anomalous the jumps are compared to those in other lakes, are significantly higher in Lakes 3 and 4 (Table 1).

All other lakes undergo changes in backscatter that are comparable with those in nearby lakes, or they experience large anomalous sudden backscatter changes but that are not sustained. Figure 5 shows the mean backscatter of Lake 6 over time together with that for the 10 largest lakes in its immediate vicinity (within a 20 km x 20 km square, centered on Lake 6). The sudden increase in mean backscatter of Lake 6 is far greater than that for the surrounding lakes. Lake 6 initially has low backscatter that is comparable with that for some of the surrounding lakes. Optical imagery from the end of the previous summer shows Lake 6 and these other 'low backscatter lakes' were water filled. Over a single image transition (12 days), Lake 6 experiences a backscatter increase to levels that are comparable with other surrounding lakes that optical imagery from the end of the previous summer showed were drained. The lakes surrounding Lake 6 experience much slower backscatter increases over time, which we interpret to be slow freezing of the water in the filled lakes or the ice surface in the bottom of the drained lakes. Figure 5 also illustrates what the backscatter changes look like within the Sentinel-1 imagery. Small changes are observable within the surrounding lakes but a much bigger change is seen in Lake 6.

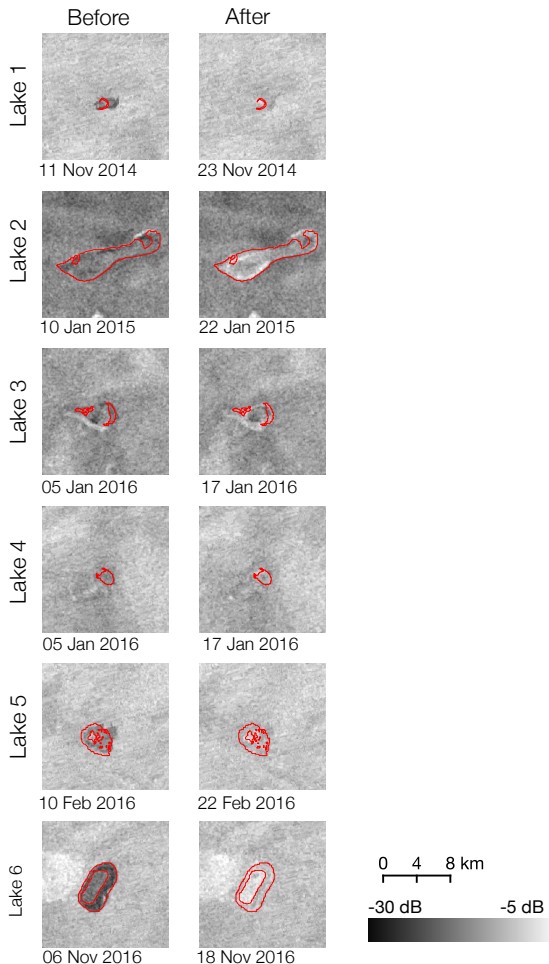

**Figure 3.** Sentinel-1 backscatter for each lake immediately before and after drainage. Before and after drainage dates are listed in Table 1. Note the lakes before drainage have a lower backscatter that changes to a higher backscatter across the image pair. Red outlines indicate the delineated lake boundary based on the NDWI$_{ice}$ threshold.

### 3.2 Confirmation of winter lake drainage by optical imagery

Analysis of Landsat-8 imagery from the summers prior and subsequent to the six inferred winter drainage events supports the interpretation that the changing SAR backscatter represents lake drainage. Using the same method described above for creating composite NDWI$_{ice}$ masks for late summer (from late July and August images), here we create similar NDWI$_{ice}$ masks for each summer but using all cloud-free Landsat-8 images between May and August from 2014 to 2017. The purpose of this is to calculate maximum lake areas for all lakes, including the six lakes inferred to drain during the winter, in the

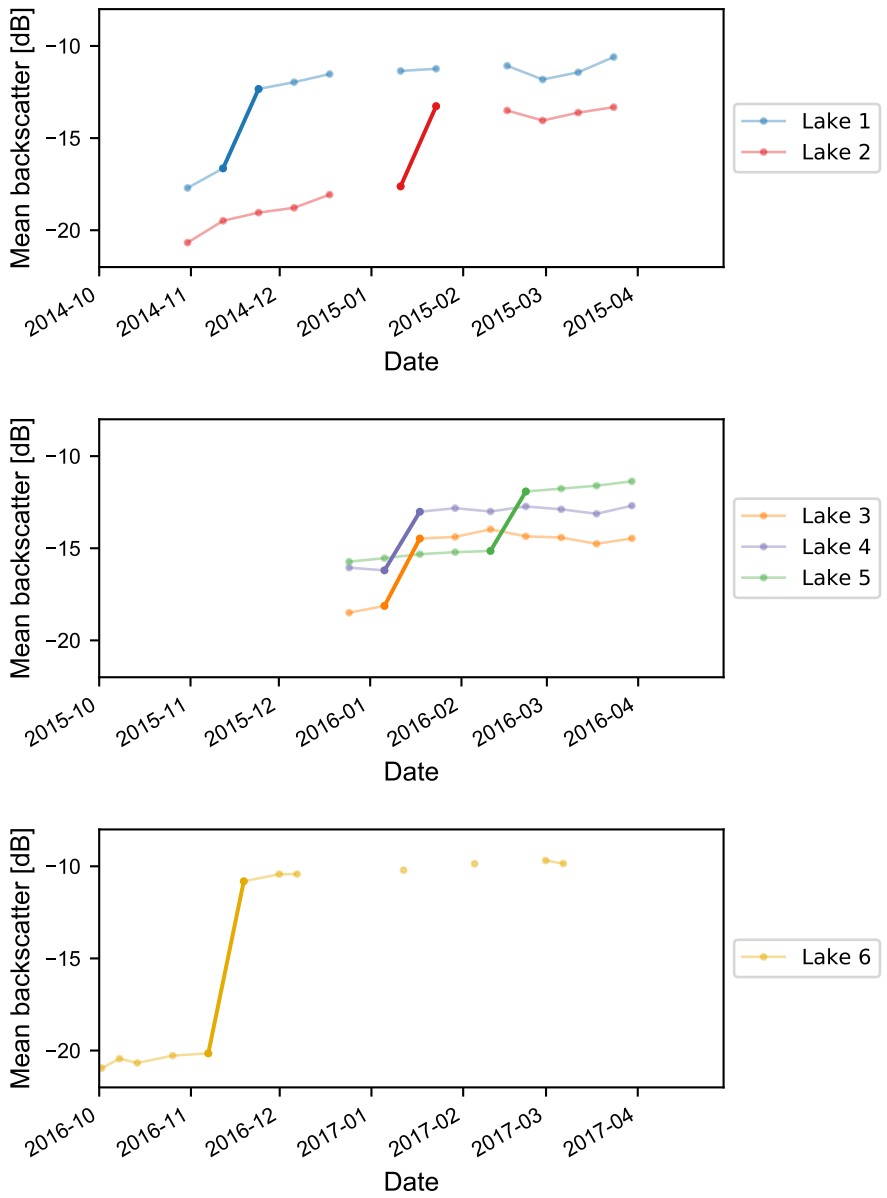

**Figure 4.** Backscatter time series for the lakes with identified drainage events. Connecting lines are only included when the time between images is 12 days or less. Each series represents one lake and each point represents the mean backscatter of all of the lake's pixels in a particular Sentinel-1 image. Bold lines indicate the transition determined to be the drainage event.

summers prior and subsequent to the winter lake drainages. Maximum summer water coverages for the six winter draining lakes are shown in Table 2. The corresponding composite NDWI$_{\text{ice}}$ images for each summer are shown in Figure 6.

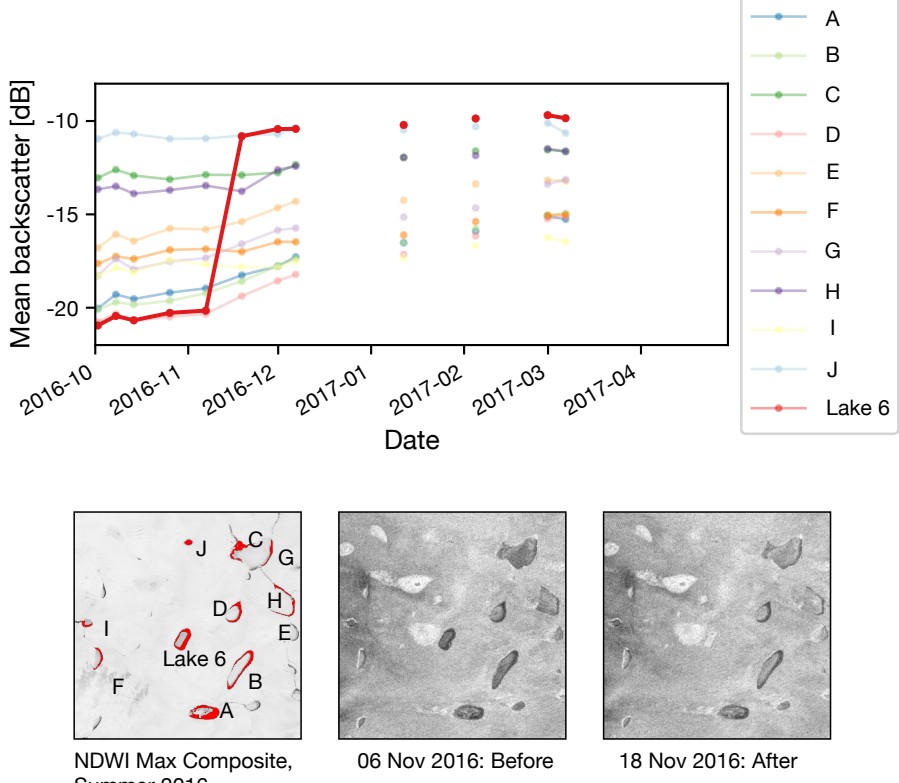

**Figure 5.** Sentinel-1 backscatter time series for the largest 10 lakes within 10 km of Lake 6. Connective lines are omitted from the time series graph when the time between images is greater than 12 days. Image (a) is a composite maximum NDWI$_{ice}$ image for late summer 2016, prior to lake drainage showing the lakes included in the graph above. Images (b) and (c) are Sentinel-1 backscatter images for 06 November 2016 and 18 November 2016 across which the drainage of Lake 6 is observed. While the backscatter of the surrounding lakes undergoes a small gradual increase over time, the backscatter increase of Lake 6 is much greater than that seen in the other lakes.

The maximum lake extents for Lakes 1, 2, 5 and 6, appear larger in the summers prior to drainage than after drainage. This
suggests that the winter lake drainages were associated with fractures / moulins that remained open, allowing the following summers' meltwater reaching the basin to drain directly into the ice sheet. These reductions in maximum lake extents contrast with those observed for the many surrounding lakes, which fill to around the same size in the adjacent summers. Lakes 3 and 4 show little difference in area before and after drainage, but the lakes do change shape (Figure 6). This suggests that the fractures / moulins associated with the winter drainage of these lakes closed shut or were advected out of the lake basins, allowing the
lakes to form again in the subsequent summer. Lakes that experience large area changes recover their area over time, but not necessarily within the first summer following drainage.

**Table 2.** Maximum lake area for each summer generated by calculating maximum NDWI$_{ice}$ per pixel from May through August each year. The lake NDWI$_{ice}$ threshold is set at 0.25 and area is calculated based on all pixels in the lake above this value.

| | Lake Areas (km$^2$) | | | |
|---|---|---|---|---|
| Lake | Summer 2014 | Summer 2015 | Summer 2016 | Summer 2017 |
| Lake 1 | 0.0936* | 0.0189 | 0.4734 | 0 (cloud cover) |
| Lake 2 | 6.498* | 0.936 | 2.774 | 3.595 |
| Lake 3 | 0.967 | 0.934* | 1.532 | 0.698 |
| Lake 4 | 0.699 | 0.639* | 0.658 | 0.495 |
| Lake 5 | 0.166 | 2.201* | 0.471 | 0 (cloud cover) |
| Lake 6 | 1.001 | 1.987 | 2.757* | 0.614 |

* indicates pre-drainage area.

## 3.3 Confirmation of lake drainage by photoclinometry and ArcticDEM differencing

Finally, we used two additional techniques to support the conclusion that the observed changes in Sentinel-1 backscatter are lake drainages. First, we used photoclinometry based on the 5 m ArcticDEM mosaic and Landsat-8 imagery before and after
255 the winter drainage events (see Methods) to calculate surface elevation changes across three of the lakes (Figure 7). Landsat-8 images suggest a smooth flat surface to each lake prior to drainage and a rough topography following drainage, suggesting the caving in of a frozen, snow-covered lake surface during drainage. Mean elevation changes calculated from photoclinometry using these images are 7.25 ± 1.61 m for Lake 2, 1.21 ± 1.61 m for Lake 5, and 3.38 ± 1.61 m for Lake 6. These depths are greater than those calculated based on the last available optical image, seen in Table 1 but are internally consistent in their rank
from smallest to largest. Possible reasons for the discrepancy between attenuation-based depth estimates and photoclinometry-based collapse depths are addressed in the Discussion.

Second, we examined differences in ArcticDEMs from dates during the winter on either side of the Lake 6 drainage event. Elevation change between time-stamped ArcticDEM strips from 21 September 2016 and 12 March 2017 is shown in Figure 8. Elevation change is greater within the lake area than surrounding it. Delineating lakes based on optically visible water means
that the lake outlines may omit possible subsurface water obscured by an ice lid. It appears from the Sentinel-1 imagery (Figure 5 and Figure 3) that Lake 6 contains a floating ice island obscuring water beneath. The mean of the differenced ArcticDEM within the NDWI$_{ice}$-based mask outline of Lake 6, is 2.17 ± 0.28 m. Note this compares with the mean depth derived from the optically-based depth calculations of 1.41 ± 0.46 m and that from the photoclinometry method of 3.38 ± 1.61 m (Figure 7). If the entire closed volume of Lake 6 is considered and the data for the entire area included in the analysis, the mean elevation
difference from the ArcticDEM strips is 3.66 ± 0.28 m and that from the photoclinometry is 4.04 ± 1.61 m.

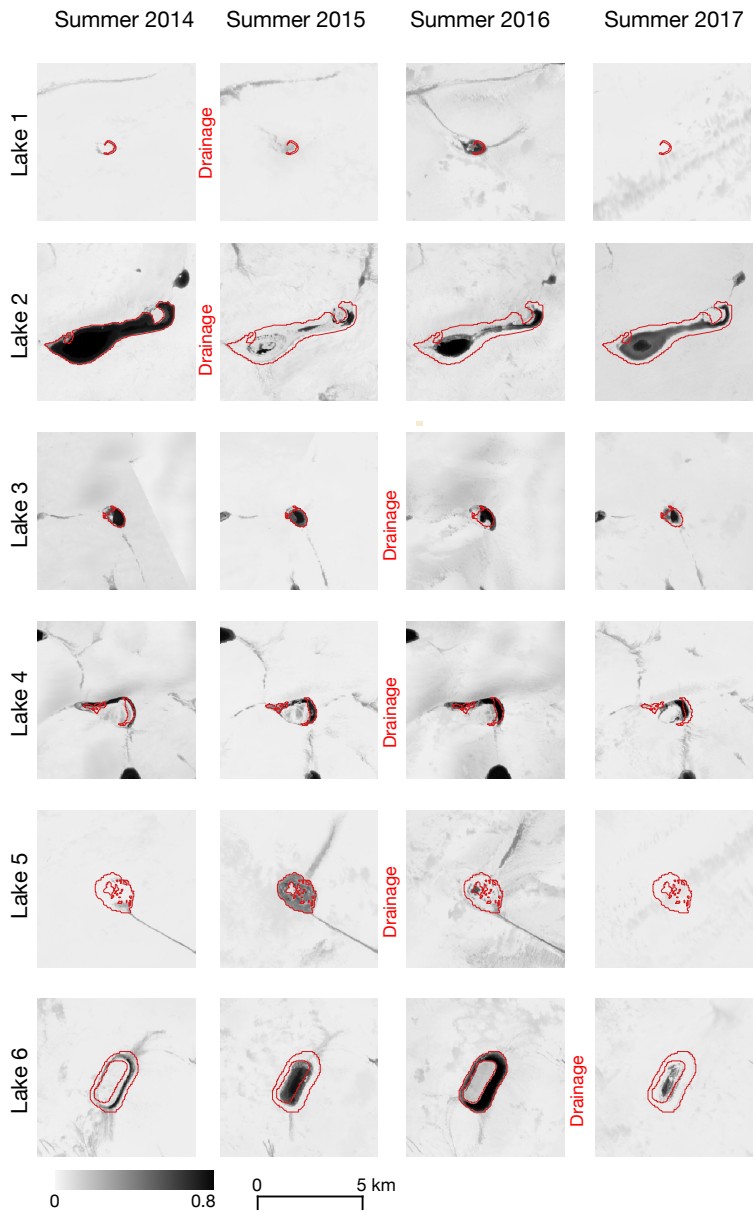

**Figure 6.** NDWI$_{ice}$ for each identified drained lake at the peak of each summer within the study. Note that most lakes take more than a single summer season to recover from their winter drainage.

## 4  Discussion

We have developed a novel algorithm for analysis of Sentinel-1 SAR imagery and used it to identify six winter lake drainage events on the GrIS, the first such events to be reported in full. Because SAR backscatter is often difficult to interpret (White

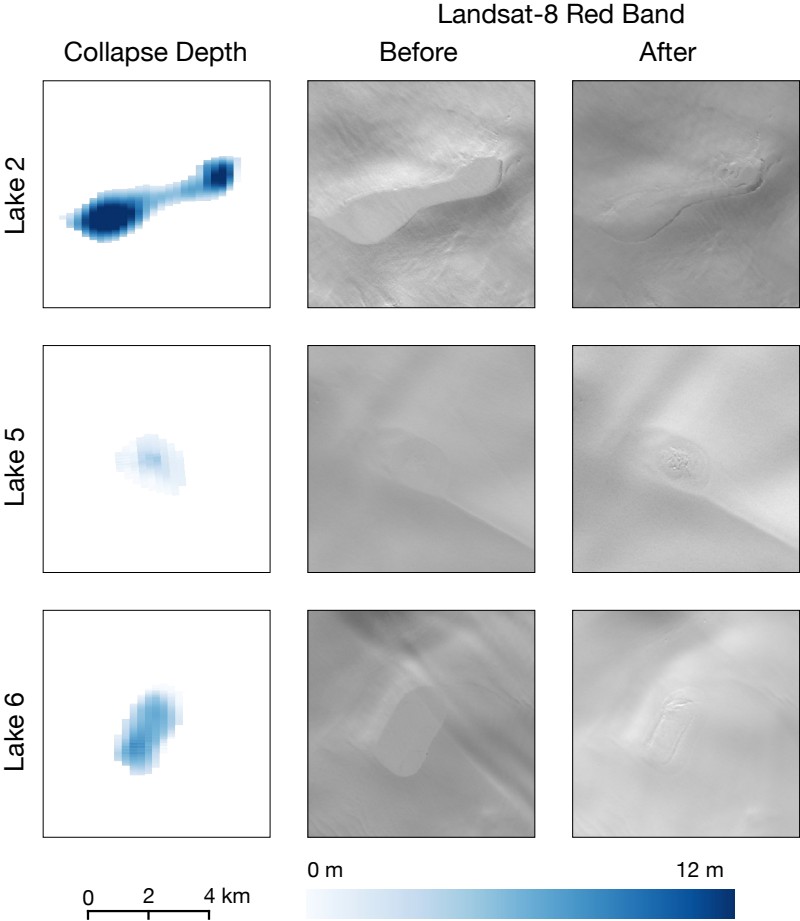

**Figure 7.** Elevation difference results of the photoclinometry analysis beside the before and after images (Landsat-8 Red Band, B4) to illustrate the visible physical changes to the lake lid before and after drainage. The first column of images shows the vertical elevation drop of each pixel calculated by interpolating and differencing the pre- and post-drainage topography.

et al., 2015) we have validated our technique by examining Landsat-8 optical imagery from the previous and subsequent summers. Changes in lake area and volume as well as topographic changes calculated using photoclinometry support the inference that these large, anomalous, sudden and sustained backscatter increases are lake drainage events. We have also been able to validate the winter drainage of one of these lakes by differencing available ArcticDEM strips.

### 4.1 Identifying lake drainage events

Identification of winter lake drainage events using Sentinel-1 data required multiple steps to isolate drainage events from other changes in backscatter. The drainage events identified occurred in lakes of various sizes and locations. If lakes are identified as

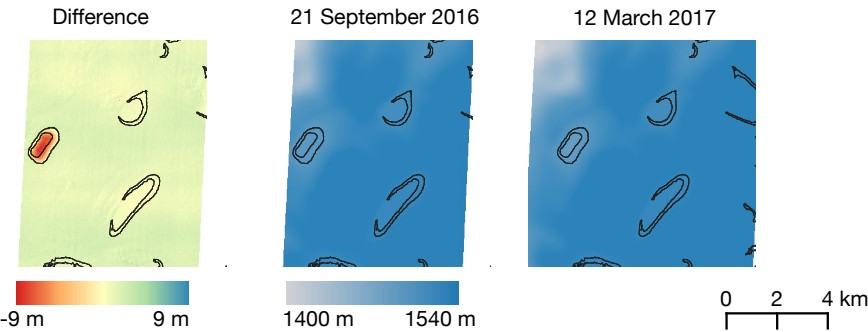

**Figure 8.** Elevation difference results of the ArcticDEM analysis to confirm the changes observed in the Sentinel-1 imagery and photoclinometry analyses.

anomalous based on z-score with no additional filtration done to confirm sustained change, the three seasons analyzed would result in 188, 160, and 221 anomalous lakes for the 2014-2015, 2015-2016, and 2016-2107 winter seasons respectively. For each of these years, retaining only lakes that met the 1.5 z-score threshold and demonstrated no reversal of trend in the first timestep would result in 75, 60, and 85 lakes, respectively. Reversal was considered to be any change greater than 25% of the

magnitude of the anomalous transition occurring either in the previous timestep or in the following three timesteps. Raising this threshold to 30% would result in 4 anomalous lakes for each season. Raising the same threshold to 40% would result in 10, 7, and 10 lakes for the three seasons, respectively. Raising it again to 50% would result in 25, 19, and 21 lakes for the three seasons.

Extending the requirement for stability by requiring more consecutive images without reversal would be difficult in most

290   years due to the limited image acquisition over this site. Overall the filtration proved not to be overly sensitive to z-score threshold, as all drained lakes had z-scores over 2 even though the threshold was set to 1.5. The criteria used to determine lake drainage events is thought to be conservative and is more likely to have missed drainage events (included false negatives) than to have found drainage events that were not real (incorporated false-positives).

### 4.2   Optical lake mask

As lake delineation using Sentinel-1 backscatter alone is not trivial (Miles et al., 2017; Wangchuk et al., 2019), all change tracking in this study is based on pixels within lake outlines generated from Landsat-8 optical imagery. However, in comparing the optically-generated masks to the Sentinel-1 backscatter images, the two are often different, typically with the SAR images showing larger lake areas than those seen in the optical data. This discrepancy may be due to water depths insufficient to meet the $NDWI_{ice}$ threshold set, or to shallow subsurface water below a snow or ice lid. This is most apparent in Lake 6 (Figures 5

and 6), where a low-$NDWI_{ice}$ island appears in the center of the lake, but HV backscatter measurements, which are sensitive to volumetric scattering, remain low in this portion and both photoclinometry and ArcticDEM changes show a caving-in of ice

in this area (Figure 5 and Figure 8). Beginning with the NDWI$_{ice}$ mask also results in the splitting of some lakes into multiple disconnected water bodies where parts of the lake are below the threshold. As such, some larger lakes may be filtered out of the study as they appear to be a collection of smaller lakes, and some backscatter tracking is only occurring on partial lakes, where only deeper portions with higher NDWI$_{ice}$ values are included in the lake delineation. Other surface changes, such as the drainage of a subglacial lake, could result in SAR backscatter changes as well. The aim of restricting the analysis to lakes that are optically-identifiable in the summer is to reduce the likelihood that the changes identified in this study are due to such events.

We have used masks created from just a few summer images to reduce the likelihood of incorporating lakes that drained in the summer into our wintertime lake tracking algorithm. Creating lake masks using a longer time span of images might allow for more complete lake boundaries to be included. By including more summer images, these masks might account for areas of water that are only occasionally seen at the surface but are more often under snow or ice, so especially those at higher elevations. Lake 1, for example, often appears below the 0.25 NDWI$_{ice}$ threshold due to the absence of cloud-free and unfrozen images within a given summer, although the lower backscatter in this area seems to indicate shallow subsurface water.

## 4.3    Sentinel-1 backscatter

While Sentinel-1 backscatter allows for the tracking of lakes that are obscured by cloud cover and darkness, it is also limited in what it can observe. The penetration depth of C-band radar producing backscatter varies based on the physical properties of the medium through which it passes, especially moisture content, but reaches a maximum of a few metres of depth (Rignot et al., 2001). However, it is also possible that winter lakes exist below this depth and are not detected. This penetration depth is also likely to be insufficient to reach the buried firn aquifers identified in the Greenland Ice Sheet (Forster et al., 2014; Koenig et al., 2014).

In our study, three images showed large, scene-wide departures from typical backscatter values and were omitted from further analysis (dated: 03 Feb 2015, 10 Apr 2016, and 16 May 2016). If it were known what caused this phenomenon then perhaps the images could be corrected and used.

Sentinel-1 is also limited in its temporal frequency of available imagery for the same site. While the repeat pass time of Sentinel-1 is at best 6 days when both satellites are included (only available since late 2016), it is advisable to use imagery from the same relative orbit for greater consistency from image to image, and not all images within each path are acquired. A shorter repeat pass could help more accurately assess the rate of backscatter change and thus gain a better understanding of the speed and timing of these drainage events. For example, no image in Relative Orbit 17 exists between 06 Nov 2016 and 18 Nov 2016, a 12 day gap in sensing, the dates between which Lake 6 drained. If additional orbits had been included in this analysis, the gap could have been reduced to 10 days, but no further.

## 4.4    Drainage water volume

Sentinel-1 backscatter alone does not allow for the calculation of water volumes and therefore water volume changes. Available optical satellite data can be used to estimate water volume, but the optical measurements are limited in their capability to

calculate accurately the drained volume. In this study, physically based depth estimates are made on a per pixel basis for each
lake using the last available image in the summer before the lake is covered by a frozen lid (Table 1). However, there are several
sources of error associated with these measurements. First, the measurements have been shown to underestimate the depth of
deep water (Pope et al., 2016; Williamson et al., 2018). Second, the measurements made months prior to the drainage events,
and the lake volumes derived from them could be impacted by additional melt filling the lake or freezing of water prior to the
drainage event. Third, the lake boundary is set using an $NDWI_{ice}$ threshold of 0.25, which may underestimate the full extent of
the lake area. Fourth, the calculation assumes all the lake water from the previous autumn drains. There is no reliable method
of using optical data to measure whether any water remains at the start of the subsequent melt season. Images showing the first
water visible in the spring after drainage could be showing water remaining in the lake or water transported into the basin from
higher elevations that year. Often cloud-free images are not available until well into the melt season and thus cannot reliably
be used as a lower bound in a calculation of water volume difference from the previous autumn.

Photoclinometry results show, for each lake, a topographical change in the surface shape between the pre- and post-drainage
images indicating an elevation drop. However, the depth of caving is greater than the deepest water depth determined from the
light attenuation based method using optical imagery from the previous autumn. The depth estimation differences may be the
result of a combination of factors. As mentioned above, the attenuation-based algorithm is known to underestimate lake depths
as the depths increase beyond a certain threshold (Pope et al., 2016; Williamson et al., 2018). Furthermore, photoclinometry-
based depths may overestimate collapse depths due to topography changes between the date of the DEM and the date of the
optical imagery used to create the shape/shading relationship. Finally, shadows within the lake basin that do not appear in parts
of the image surrounding the lake may also introduce errors into the calculation of shape from shading within the basin.

While the depth estimation using this photoclinometry may be inaccurate in places for the reasons outlined above, the
technique confirms that a change in surface topography occurred. Photoclinometry is potentially a useful method for detecting
surface or shallow subsurface lake drainages on ice sheets and ice shelves. The optical data support the assertion that the
changes in winter SAR backscatter observed are caused by lake drainage events. The larger lakes in the study, Lakes 2, 5, and
6 all show a significant reduction in lake area in the summer following the winter drainage compared to the previous summer
with more than a single summer season needed to regain pre-drainage lake area (Figure 7). This may be due to the opening
of a fracture that continues to allow water to drain through the lake bed for some time, similar to that found by Chudley et al.
(2019). Lake 1 shows a similar slow re-filling over time but the effect is less clear in Lakes 3 and 4.

Compared to Lakes 1, 2, 5, and 6, Lakes 3 and 4 did re-fill to their former size in the summer following drainage (Figure 6).
While these two lakes did show a large, anomalous sudden and sustained backscatter increase suggesting winter lake drainage
according to our criteria, they were small in area and the subsequent filling makes it less clear that drainage events actually
occurred. These lakes also lack the additional support of photoclinometry or ArcticDEM differencing that the lakes definitively
drained. The SAR backscatter changes suggest that the lakes did drain, and if this is the case, the available optical data suggest
that any fracture created during drainage may have been subsequently squeezed shut or advected out of the small lake basin
allowing the lakes to fill again the following summer.

The drainage of Lake 6 is further confirmed by the analysis of the ArcticDEM differential (Figure 8), which shows a collapse across the entire lake area, including the central area that did not appear as deep water in any preceding-summer Landsat-8 images. The collapse is greatest at the center and decreases toward the edges of the lake boundary. The magnitude of the collapse as measured by the DEM differential is similar to that measured by the photoclinometry method. Furthermore, the nearby lakes show no significant elevation change across the same period.

## 4.5 Causes and implications of lake drainage

The causes of lake drainage events have been studied extensively (Williamson et al., 2018; Christoffersen et al., 2018). However the observation of isolated winter lake drainages points to the possibility that drainages can occur without increases to lake volume to actively cause hydrofracture or to connect to a nearby moulin to trigger sliding or uplift and passively open a crack. Instead, it shows that ice dynamics unrelated to surface hydrology can trigger drainage. The evidence available in this study is insufficient to identify conclusively the cause of these winter lake drainages. Appendix Figure B1 shows the locations of the winter lake drainage events compared to ice speeds derived from MEaSUREs data (Howat, 2017) for the winter periods containing each drainage event. There is no obvious correlation between ice speed patterns and the location of winter lake drainage events suggesting that patterns of ice flow are not necessarily a trigger for drainage. Our sample size is small, however, and more evidence is needed to examine further the possibility. In this study, most of the lake drainages occur in isolation - with the exception of the drainages of Lakes 3 and 4, which occur in the same 12-day period. These lakes are separated by a linear distance of 14.9 km. These concurrent drainage events may be related, with one drainage triggering the other by creating localised ice acceleration transferred via stress gradients (Christoffersen et al., 2018). Alternatively, they may indicate a larger scale ice movement that triggered both events simultaneously.

## 5 Conclusions

We have developed an automated method for identifying large, anomalous, sudden and sustained backscatter changes in Sentinel-1 SAR imagery, which we apply to images collected between October and May spanning three winter seasons. We find four winter lake drainage events across a study site containing approximately 300 supraglacial lakes that are supported by optical data and two other possible drainage events that meet our backscatter change criteria but lack the optical data support to unequivocally confirm drainage.

The optical imagery from before the winter seasons are used to provide estimates of lake volumes associated with the drainages. While the events are rare, they provide conclusive evidence for the first time that lake drainages over winter occur. They are likely triggered simply by crevasse opening across the lake due to high surface strain rates associated with background winter ice movement. This shows that rapid lake drainage events do not have to be triggered during lake water filling, as has been observed previously for summer events. A full picture of the hydrology of the Greenland Ice Sheet requires observation of surface water on a multi-year and multi-season basis. Identification of the drainage events was achieved by developing a time-series filtering algorithm that may be adapted to identify other hydrological phenomena such as the onset of melt, or the

rate of filling or freezing of surface or shallow subsurface water bodies on ice sheets and ice shelves. The algorithm is based on a set of thresholds that were set conservatively to capture only the most obvious incidences of large, anomalous, sudden and sustained backscatter changes and therefore our study is more likely to have underestimated rather than overestimated the number of winter lake drainages (included false negatives rather than false positives). Further work is required to examine whether winter lake drainage occurs in other parts of the ice sheet and in other years, what the triggering mechanisms are, how basal hydrology and biogeochemistry are affected, and whether winter lake drainage will become more prevalent under future climate warming scenarios.

*Data availability.* All data used in this study are available publicly through ESA, USGS, and Google Earth Engine.

## Appendix A: Appendix A: Photoclinometry process

### A1 List of Landsat-8 images used for Photoclinometry

| Lake | Landsat-8 Scene |
| --- | --- |
| Lake 2 Before | LC08 _008012 _20141101 |
| Lake 2 After | LC08 _008012 _20150221 |
| Lake 5 Before | LC08 _008011 _20151104 |
| Lake 5 After | LC08 _008011 _20160428 |
| Lake 6 Before | LC08 _009011 _20161028 |
| Lake 6 After | LC08 _009011 _20170217 |

### A2 Slope vs. Reflectance

Figure A1 shows the correlation of slope with reflectance for the non-lake areas of each of the Landsat-8 images used in the photoclinometry section of this study. For each image, a new relationship was established and used to infer the slope of the lake area within that image.

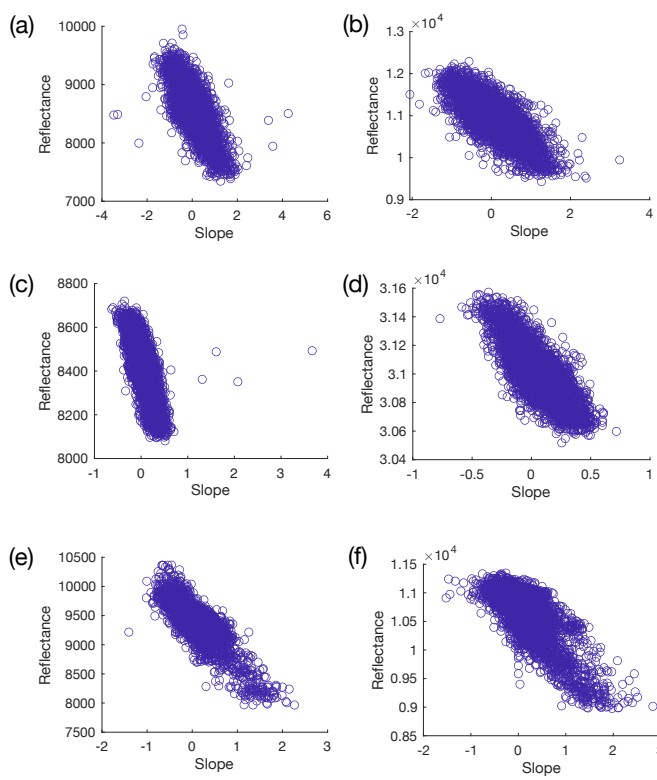

**Figure A1.** Plots of slope vs. Landsat-8 red band reflectance for areas outside of the lake and buffer zone for each of the Landsat-8 images analyzed for the photoclinometry portion of this study. The plots are laid out as follows: (a) Lake 2 Before, (b) Lake 2 After, (c) Lake 5 Before, (d) Lake 5 after, (e) Lake 6 Before, and (f) Lake 6 After.

**A3   Lake sampling**

Figure A2 shows the set up for the photoclinometry portion of the study. The lake was manually outlined and buffered, and transects were spaced every 250 m and sampled every 30m along transect for each 10 km long transect.

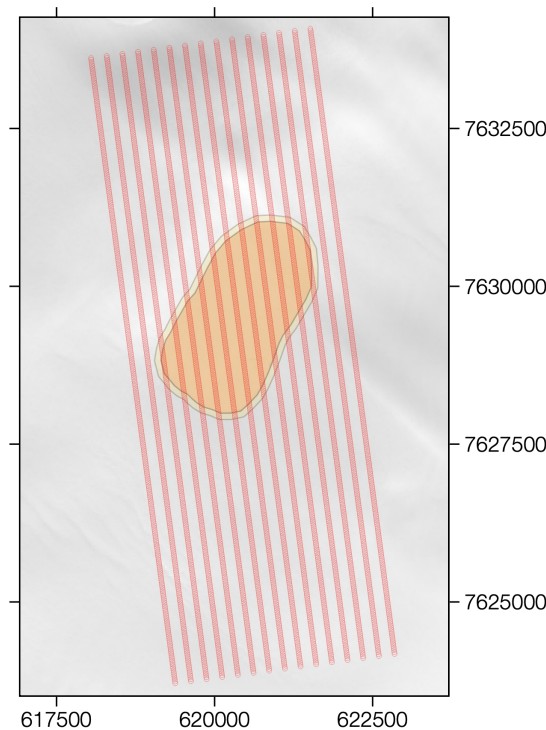

**Figure A2.** Lake 6 transects for photoclinometry calculations for image on 28 Oct 2016 prior to drainage (red), lake extent (orange) and buffer (yellow). For description of how these features are used in the photoclinometry calculations, see Methods.

## A4   Photoclinometry point sampling

Figure A3 shows the correction of a transect across the lake. Transect 'A' in the graph was the original transect calculated following the photoclinometry process. Transect 'B' is the result of correction by calculating the elevation difference between the end of the lake transect and the elevation at that lake edge in the ArcticDEM and then distributing that elevation difference evenly across the lake transect.

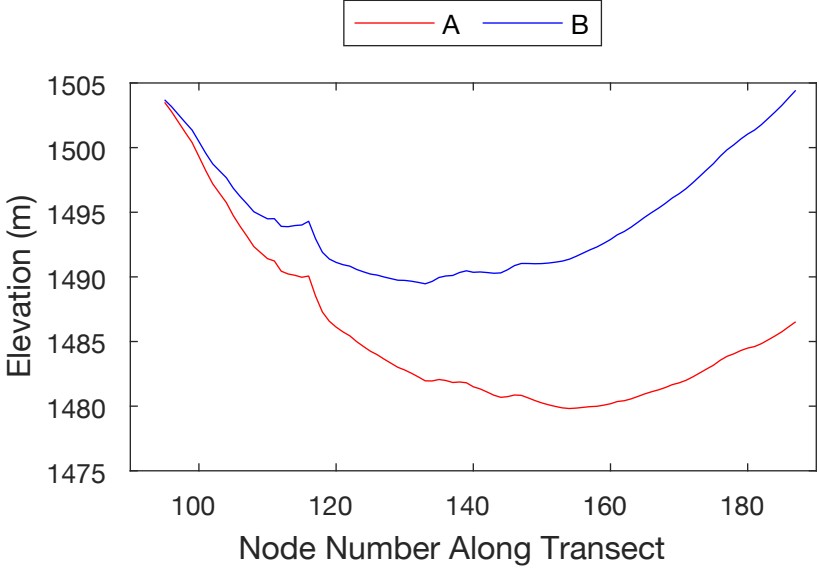

**Figure A3.** An example Lake 6 transect pair for photoclinometry calculations before (red), and after (blue) correction.

## Appendix B: Appendix B

Figure B1 presents pixel by pixel ice speeds based on MEaSUREs velocity data (Howat, 2017) for the winters surrounding
each of the drainage events.

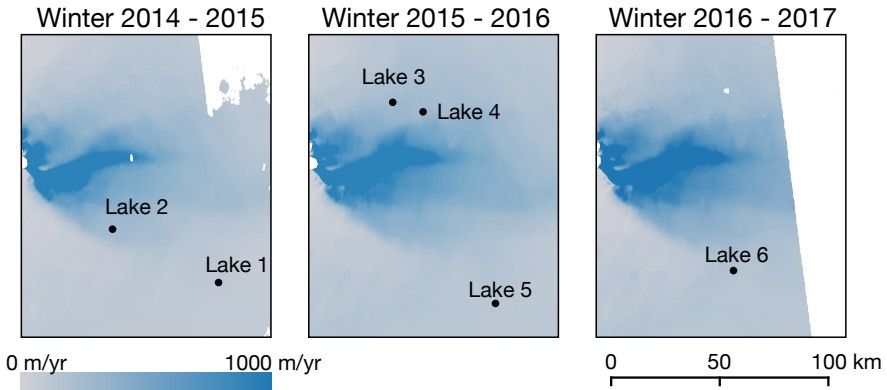

**Figure B1.** Ice speeds for the winter quarter proximate to each of the lake drainages.

## Appendix C: Appendix C

This figure shows the behaviour of lakes surrounding the identified drained lakes for the summers included in this study. The images shown are peak values of $NDWI_{ice}$ for each pixel, creating a maximum composite image. Red shading covers the extent of the lake mask for each year.

## 430   Appendix D:  Appendix D

The figures in this appendix show the backscatter time series for the lakes proximate to each of the identified drainage events. These are the equivalent of Figure 5 for all the lakes apart from Lake 6, which is shown in the body of the paper. For each figure, the top panels shows the backscatter time series. In the bottom row, (a) shows the lakes captured by the $\text{NDWI}_{\text{ice}}$ mask, (b) shows the backscatter prior to the drainage event and (c) shows the backscatter after the drainage event.

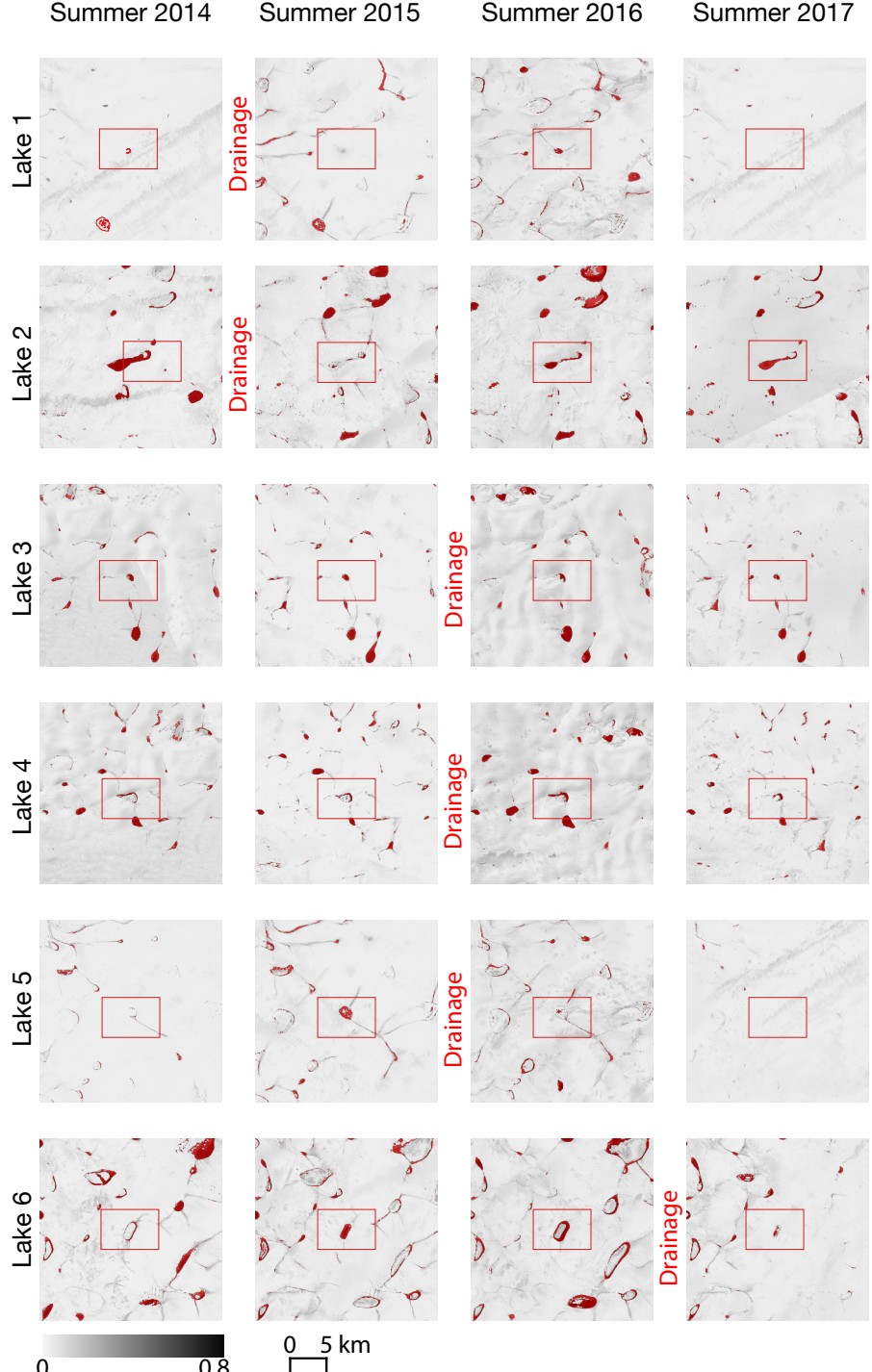

**Figure C1.** Composite maximum NDWI$_{ice}$ images for each summer. Each pixel shows the highest NDWI$_{ice}$ reached for that pixel for the season. The red outlines show the lake outlines as delineated by a threshold exceeding 0.25 in the maximum NDWI$_{ice}$ composite for the pre-drainage summer. Red boxes identify each anomalous lake.

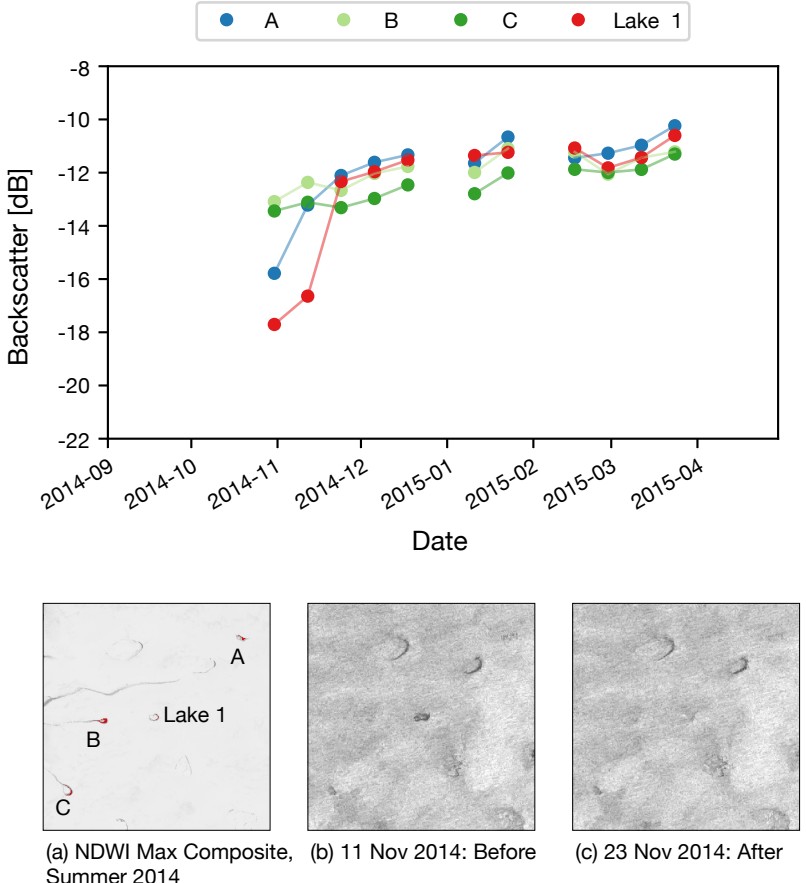

(a) NDWI Max Composite, Summer 2014

(b) 11 Nov 2014: Before

(c) 23 Nov 2014: After

**Figure D1.** Lake 1 Surrounding Lakes

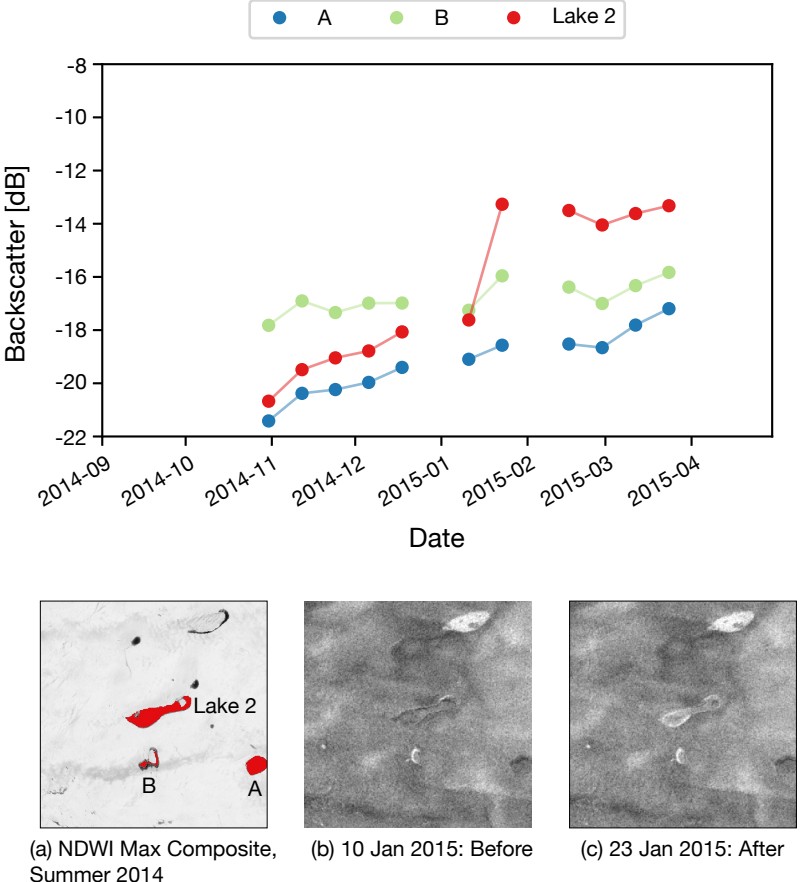

(a) NDWI Max Composite, Summer 2014

(b) 10 Jan 2015: Before

(c) 23 Jan 2015: After

**Figure D2.** Lake 2 Surrounding Lakes

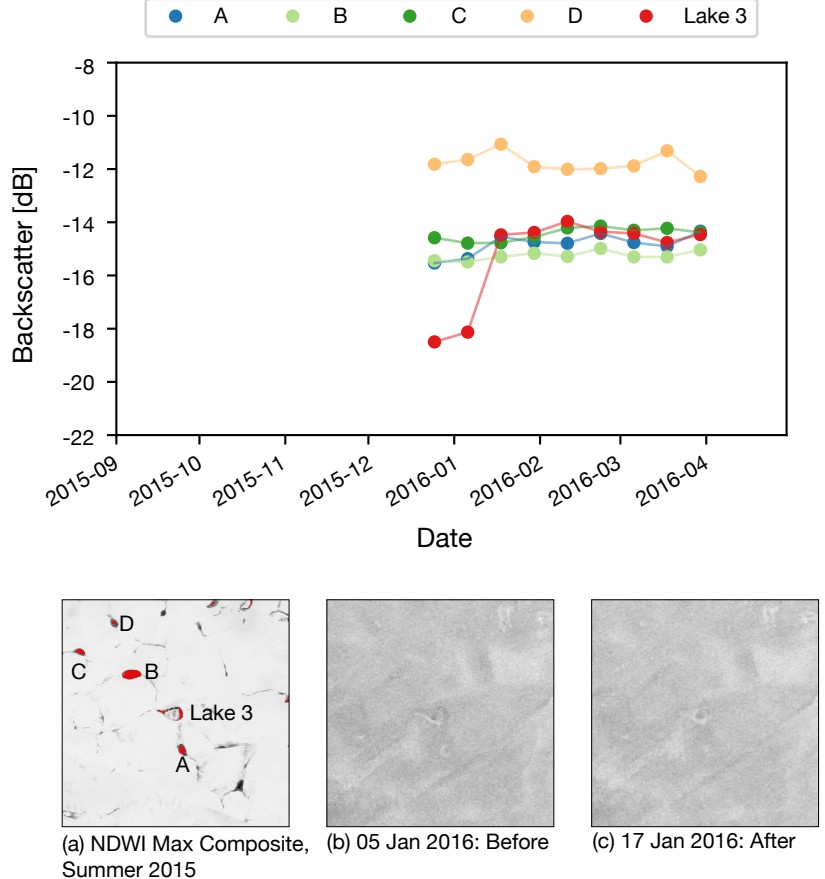

(a) NDWI Max Composite, Summer 2015

(b) 05 Jan 2016: Before

(c) 17 Jan 2016: After

**Figure D3.** Lake 3 Surrounding Lakes

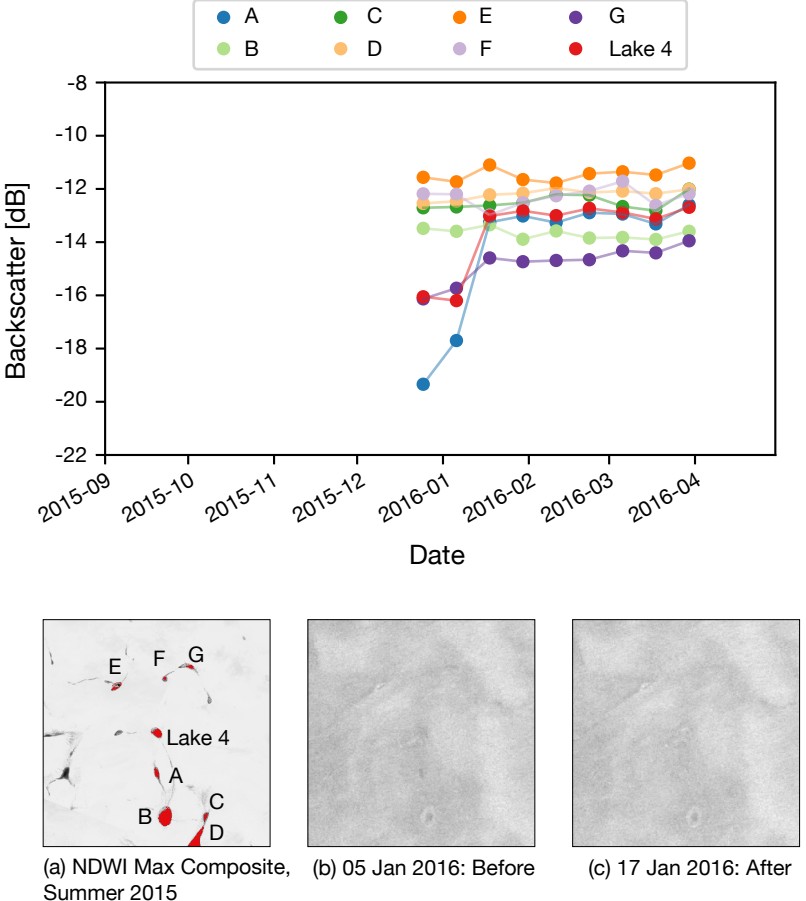

(a) NDWI Max Composite, Summer 2015

(b) 05 Jan 2016: Before

(c) 17 Jan 2016: After

**Figure D4.** Lake 4 Surrounding Lakes

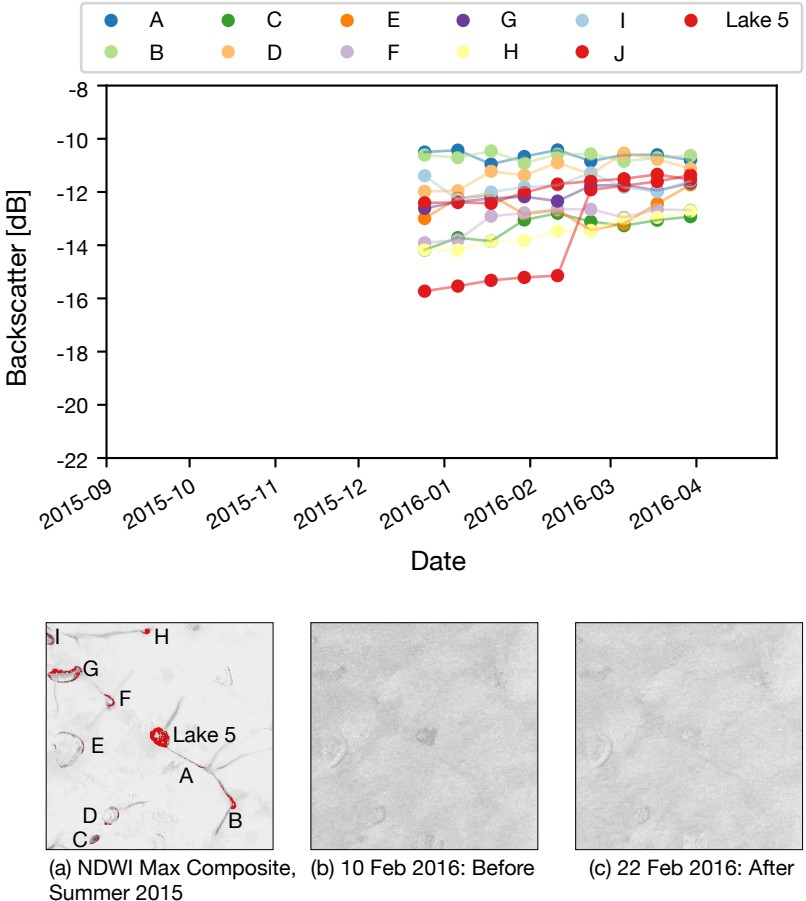

(a) NDWI Max Composite, Summer 2015

(b) 10 Feb 2016: Before

(c) 22 Feb 2016: After

**Figure D5.** Lake 5 Surrounding Lakes

## Appendix E: Appendix E

The following is a list of Landsat-8 image IDs included in the late-summer NDWI$_{ice}$ max composite images used to delineate the lake boundaries for backscatter analysis.

Late Summer 2014 Max Composite

LC08_006013_20140815

LC08_007012_20140806

LC08_007013_20140806

LC08_007013_20140822

LC08_008011_20140829

LC08_008012_20140829

LC08_008013_20140813

LC08_008013_20140829

LC08_009010_20140804

LC08_009011_20140804

LC08_009012_20140804

LC08_009013_20140804

LC08_010010_20140811

LC08_010010_20140827

LC08_010011_20140811

LC08_010011_20140827

LC08_010012_20140811

LC08_010012_20140827

LC08_010013_20140811

LC08_010013_20140827

LC08_011009_20140802

LC08_011010_20140802

LC08_011010_20140818

LC08_011011_20140802

LC08_011011_20140818

LC08_011012_20140802

LC08_011012_20140818

LC08_012009_20140825

LC08_012010_20140825

LC08_012011_20140825

LC08_014009_20140807

LC08_014009_20140823

LC08_014010_20140807

LC08_014010_20140823

LC08_016009_20140805

LC08_016009_20140821

LC08_082234_20140804

LC08_082235_20140804

LC08_084234_20140802

Late Summer 2015 Max Composite

LC08_006013_20150802

LC08_006013_20150818

LC08_008011_20150731

LC08_008011_20150816

LC08_008012_20150731

LC08_008012_20150816

LC08_008013_20150731

LC08_008013_20150816

LC08_009011_20150807

LC08_009013_20150807

LC08_010010_20150729

LC08_010010_20150814

LC08_010011_20150729

LC08_010011_20150814

LC08_010012_20150729

LC08_010012_20150814

LC08_010013_20150729

LC08_010013_20150814

LC08_011009_20150805

LC08_011010_20150805

LC08_011011_20150805

LC08_011012_20150805

LC08_012009_20150727

LC08_012010_20150727

LC08_012011_20150727

LC08_012011_20150812

LC08_013009_20150803

LC08_013009_20150819

LC08_013010_20150803

LC08_013010_20150819

LC08_013011_20150803

LC08_013011_20150819

LC08_014009_20150725

LC08_014009_20150810

LC08_014010_20150810

LC08_015009_20150801

LC08_015009_20150817

LC08_015010_20150801

Late Summer 2016 Max Composite

LC08_006013_20160804

LC08_006013_20160820

LC08_008011_20160802

LC08_008011_20160818

LC08_008012_20160802

LC08_008012_20160818

LC08_008013_20160802

LC08_008013_20160818

LC08_009010_20160809

LC08_009011_20160809

LC08_009012_20160809

LC08_009013_20160809

LC08_010010_20160816

LC08_010011_20160816

LC08_010012_20160816

LC08_010013_20160816

LC08_011009_20160807

LC08_011010_20160807

LC08_011011_20160807

LC08_011012_20160807

LC08_012009_20160814

LC08_012010_20160814

LC08_012011_20160814

LC08_013009_20160805

LC08_013009_20160821

LC08_013010_20160805

LC08_013010_20160821

LC08_013011_20160805

LC08_013011_20160821

LC08_014009_20160812

LC08_014010_20160812

LC08_015009_20160803

LC08_015010_20160803

LC08_016009_20160810

LC08_081235_20160801

LC08_082235_20160809

Late Summer 2017 Max Composite

LC08_006013_20170722

LC08_006013_20170823

LC08_007012_20170814

LC08_007013_20170814

LC08_008011_20170805

LC08_008011_20170821

LC08_008012_20170805

LC08_008012_20170821

LC08_008013_20170720

LC08_008013_20170805

LC08_008013_20170821

LC08_009010_20170812

LC08_009011_20170812

LC08_009011_20170828

LC08_009012_20170812

LC08_009012_20170828

LC08_009013_20170812

LC08_009013_20170828

LC08_010010_20170819

LC08_010011_20170819

LC08_010013_20170819

LC08_011009_20170725

LC08_011009_20170810

LC08_011009_20170826

LC08_011010_20170725

LC08_011010_20170810

LC08_011010_20170826

LC08_011011_20170725

LC08_011011_20170810

LC08_011011_20170826

LC08_011012_20170725

LC08_011012_20170826

LC08_012009_20170716

LC08_012009_20170801

LC08_012010_20170716

LC08_012010_20170801

LC08_012011_20170801

LC08_013009_20170723

LC08_013009_20170808

LC08_013009_20170824

LC08_013010_20170723

LC08_013010_20170808

LC08_013010_20170824

LC08_013011_20170723

LC08_013011_20170808

LC08_014009_20170730

LC08_014010_20170730

LC08_014010_20170815

LC08_015009_20170721

LC08_015010_20170721

LC08_016009_20170813

LC08_084233_20170725

LC08_084234_20170725

LC08_085231_20170716

LC08_085234_20170801

LC08_086233_20170723

*Author contributions.* Both authors conceived of the work, contributed to the ideas and wrote and edited the paper. CB performed all the analysis and produced all the Figures.

*Competing interests.* The authors declare no competing interests.

*Acknowledgements.* CB is funded by the Howard Research Studentship through Sidney Sussex College and the Cambridge Trust. The ArcticDEM was downloaded from Google Earth Engine through the Polar Geospatial Center, University of Minnesota. DEM(s) were created from DigitalGlobe, Inc., imagery and funded under National Science Foundation awards 1043681, 1559691, and 1542736. We thank Marco Tedesco, Neil Arnold, Gareth Rees, Tom Chudley, and Andrew Williamson for useful discussions about various aspects of this work at different stages. The authors would also like to thank Andrew Sole and one other anonymous referee for their detailed and helpful comments. Their contributions helped strengthen the paper considerably.

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
