# Peer review of "Winter drainage of surface lakes on the Greenland Ice Sheet from Sentinel-1 SAR Imagery"

_The Cryosphere, 2020_

## Referee Comment (RC1) · Andrew Sole (Referee) · 24 Jun 2020

General comments

This manuscript presents the first quantification of the drainage of supraglacial lakes in Greenland during winter. Such events have previously only been described qualitatively, or their occurrence inferred from proglacial river data. As such, the authors make a worthwhile contribution to help fill in some gaps in our understanding of ice sheet hydrology. The paper is on the whole clearly written and the data analysis is valid and suitable (barring a few inconsistencies – see specific comments below). The main conclusions are justified, although there are some overly speculative comments made at

the very end of the manuscript. My main comment is that the temporal coverage of the radar data used is limited. Sentinel-1b only started consistently retrieving data from west Greenland in October 2016, so a 6-day period for the same relative orbit is only possible from then. This raises the question of why the authors did not look for winter lake drainages over more recent years (i.e. after 2016/17). Doing so might improve the temporal resolution of the data and thus avoid some of the limitations. There are also a number of more specific points that need to be addressed.

Specific comments

L2: 'immediately' seems to contradict the 'hours to days' later in the sentence. I suggest removing it.

L3 & L26: Is meltwater access always sustained for the rest of the summer? If the ice is thick (so that creep closure rates at the base of the moulin are rapid) and surface meltwater input following lake drainage is low (i.e. the lake and moulin are at high elevation), the moulin might close and the lake refill.

L26: 'This' should be 'Drainage' otherwise it is somewhat vague what is being referred to.

L27: Not necessarily the 'down-glacier direction'. The direction of subglacial water flow is determined by the subglacial hydropotential surface, the slope and aspect of which will vary from that of the ice surface (due to the bed topography) and may be different from the broad definition of 'down-glacier'.

L32: It might be worth adding that the ice speed often decelerates below the pre drainage value because of the temporary increases in basal hydraulic efficiency.

L36: Although lakes contribute to total runoff from the ice sheet, they do not 'control' it. If you look at a seasonal hydrograph (e.g. Bartholomew et al. (2011, doi:10.1029/2011GL047063)), the overall shape is determined by atmospheric temperatures and ice surface melt rates. Because the highest melt rates are closer to

the margins at lower elevations where there are fewer lakes, most meltwater enters the subglacial drainage system via crevasses and moulins not associated with lakes (Koziol et al. 2017). Lake drainages are typically superimposed on this seasonal pattern.

L48 - 49: This last part of the sentence doesn't quite make sense to me.

L52: You should use the final TC reference which is 2013 (also in the reference list).

L64: More recent data acquisitions from Sentinel-1 a and b are more consistent and regular. Did you look over the 2017-2018 and later winters and not find any lakes? Or have you not looked at these data? Doing so might remove some of the temporal frequency limitations you mention later in the manuscript.

L90: I wonder if it is worth mentioning somewhere that subglacial lake drainage (and the resulting formation of so-called 'collapse basins') might lead to a similar change in radar backscatter. The fact that you used a supraglacial lake mask to search for the backscatter changes suggests that the changes you identified were supraglacial lake drainages, but it might be worth a mention nonetheless.

L105. The 'therefore' does not quite follow as written, but needs more explanation in the previous sentence justifying why you'd expect gradual freezing to lead to an increase in backscatter. Also, you should provide more details about why you think that a lake drainage would lead to a sudden, significant and sustained increase in backscatter. Is it because the collapsed lid of the lake would create chaotic relief and therefore be bright, or is it just the change from the radar 'seeing' through the frozen to the lake surface, to the radar instead seeing the ice of the drained lake bed?

L106: I think the comparison with a summer lake drainage is probably valid but requires a bit more explanation. In the summer case, the backscatter values change because the surface changes from water to ice. It is likely the same change that is seen in winter (even though the lake might be partially frozen over) because C-band SAR can

penetrate a few m of ice - likely thicker than the frozen lake surface, at least in the early part of the winter.

L121: It would be useful to also state the actual area in metres squared

L125: Should it not be the latest rather than the greatest? Otherwise the estimated volume might be significantly greater than it was at the time the lake drained. Later in the manuscript you do refer to the volume estimates being for the last Landsat image of the season, so I think there is a mistake somewhere here.

L133: Did the image tiles include any seawater? If so, was this used as the darkest pixel? Might the darkest pixel not be from a lake with sediment at its base and thus not truly representative of the spectral signal of deep water?

L157 – 160: Understanding of this process would be greatly aided by the addition of an explanatory diagram.

L172 – 174: But you used the Landsat image with the greatest area for the lake depth rather than the latest one (L125). It is also possible that the lake volume reduced following your Landsat-derived volume calculation.

L179: It would be useful to show the extent of the optical lake masks on the Sentinel-1 backscatter images to see over what area the mean change in dB is calculated. Also, might a median value be less prone to the influence of outliers?

L186: 'identified' would be better than 'filtered out' (otherwise it seems like you are removing them from the time series)

L186 – 187: This repeats some of the methods section really. Is it needed here again? 'All other lakes. . .' could follow logically straight on from the previous paragraph.

Figure 3 caption: Does the last sentence definitely apply to this figure? It does not seem to make sense.

L198 (subtitle 3.2): It would be useful to state in the section title what you are confirming

- 'Confirmation of winter lake drainage. . .'

L219: Average depth for Lake 2 after drainage is more than double that calculated when the lake was present. Why do you think the differences are so large? Do you only calculate the depth of the depression to the lake shoreline using photoclinometry? Apologies if I've misunderstood the method, but I found it difficult to follow.

L227: 'calculated using' might be better than 'expressed through'

Figure 7 caption: The second and third sentences are a bit convoluted. I suggest changing to: 'The first column of images shows the collapse vertical distance of each pixel calculated by interpolating and differencing the pre- and post-drainage topography.'

L232: I think it would be worth briefly reiterating how you used the z-score – I.e. the z-score of backscatter change for each lake is calculated relative to the backscatter change of all lakes across the scene

L250: C-band SAR penetrates a few m of ice (Rignot et al. 2001), so likely sees through the nascent ice lid. I think this needs to be stated more clearly early on. You discuss the low backscatter values in a somewhat vague manner initially before offering an explanation in Section 3.3.3. Perhaps it would make more sense to swap the order of Sections 3.3.2 and 3.3.3?

L257 – 258: Based on Figure 5 you might have more luck using Otsu thresholding on the Sentinel-1 images, as this would 'fill in' the interior of many of the lakes that are doughnut shaped in the NDWI composite.

L266: The value of 9 m is for dry cold firn. It will be less for the ice lids on the lakes (a few m or less I expect based on Rignot 2001).

L272: Be clear that this is temporal frequency

L273: Both satellites were only recording image consistently from c. October 2016

L282 – 283: Here you state that the depth estimates were based on the last available image, but on L125 you state that the depth measurement was based on the image when the lake was largest.

L285 – 287: Based on the above discrepancy in how you measured the lake depth, your estimate might very well be an overestimate rather than an underestimate. This needs to be cleared up and the justification of why the lake depth and the photoclinometry depth are so different amended accordingly.

L290 – 291: In terms of determining whether water was transported into the basin from higher elevations, could you not compare the dB values with the maximum achieved over the winter to detect surface melt at higher elevations? You could also use the runoff output of a regional climate model like RACMO.

L293 – 294: Have you considered using the ArcticDEM time-stamped data strips? There may be some that would help to further constrain the volume of the drained lakes. See e.g. Livingstone et al. (2019) https://doi.org/10.5194/tc-13-2789-2019

L301: Maybe remind the reader that this refers to the 5 m mosaicked product so is made up of data from many different times.

L303: Changes in backscatter are 'caused by' lake drainage events

L313: What about short sharp melt events over winter? Have you looked at any available meteorological data? Also do you detect a reduction in backscatter for the non-lake surface at the same time the lake backscatter increases? This might indicate a small amount of surface melting that might have an effect on the (presumably relatively inefficient) subglacial drainage system if it got to the ice bed.

L316: The transient nature of any speed-up probably means that there would be no discernible signal in a winter average velocity estimate.

L317 – 318: I'm not sure your sample size is big enough to be able to say this definitively, so it may be worth including this caveat.

L322 – 325: Without actually doing a rough calculation (basin or lake diameter, velocity and time) this seems overly speculative.

l319: the term 'cascade draining' is a little misleading (although I realise it is used in the title of the Christoffersen paper). Perhaps add a very brief explanation of the process – i.e. drainage of one lake creates ice acceleration and a tensile shock that is transferred through the ice and can trigger other lakes to drain etc.

L329: I don't think it is necessary to repeat 'large, sudden, anomalous and sustained' here.

L329 & 332: I think it is worth specifying that you are talking about supraglacial lakes here (for anyone who might just read the conclusion).

Technical corrections:

Figure 2: Lines need to be thicker and symbols larger (and C is very difficult to see)

L148: missing space between value and units

L151: Do you mean Appendix A? Appendix B appears to show ice velocity data.

L230: 'event' should be 'events'

L242: 'false negative ones' should be 'false negatives'

L243: 'false positive' should be 'false positives'

---

## Referee Comment (RC2) · Anonymous Referee #2 · 26 Jun 2020

General comments:

This manuscript presents evidence of 6 different winter lake drainages across the Greenland Ice Sheet. The authors use a variety of methods (Sentinel-1 backscatter tracking, optical imagery analysis, photoclinometry) to provide evidence of these lake drainages and quantify drainage volumes. The findings presented in this paper are a valuable contribution to better understanding Greenland Ice Sheet hydrology. My first concern with the paper is that the writing is, at times, hard to follow. This is particularly true within the Methods section where overly wordy sentences take too long to dissect and comprehend. My second concern is that I am not convinced by the evidence for

the 'drainages' of lakes 3 and 4 for reasons which I have further discussed below. Additionally, there is no elevation-change analysis from photoclinometry for these lakes. I understand that this may not be possible with the available Landsat-8 images; however, I don't believe that the evidence presented is convincing.

Specific comments:

L7 – specify which winters

L36 – Sentence beginning with "Lake drainage events, therefore,..." seems out of place within the rest of this paragraph.

L37 – Where do drainage events raise levels of phosphorus, nitrogen and sulfate?

L43 – I don't believe Koenig it al (2015) documented lake drainages, just the existence of winter-stored meltwater.

L44 – Perhaps combine these two sentences so the second one doesn't start with "They".

L47-49 – The sentence beginning with "conventional understanding" does not make sense

L51-53 – This sentence is somewhat unclear to me

L54 - delete "carefully" and "in" in "microwave backscatter in Sentinel-1 satellite"

L76 – what are the dates that determine a "late season" image?

L86 – Would it make more sense to use the last optical image from the summer to define the lake boundaries instead of the maximum?

L111 – What does "lakes across the scene" mean? How large of an area is this?

L112 – What does the last sentence mean?

L125-126 – Again, would it make more sense to use the last optical image from the

summer to calculate lake volumes instead of the maximum lake area?

L175 – I imagine that partial re-freeze would greatly impact the lake volume. Some water must have frozen as these lakes are no longer on the surface but are buried beneath a layer of ice. Also, I am wondering how the lake area detected from optical imagery compares with lake area detected from S1 imagery immediately prior to collapse? I image that the outlines of lake 3 and 4 would look quite different between the optical and S1 imagery.

Table 1 – What are the uncertainties on lake depth and volume?

L190 – With regards to Lake 6: I looked briefly at this lake on GEE during this time period using the HH band. I noticed that surrounding lakes show an increase in backscatter similar to lake 6 with the HH band. Do you have an explanation for this?

Figure 3 – I believe it would be useful to include dates on these images. Also the last line of the caption seems misplaced. Finally, I am not convinced by the 'drainages' of lakes 3 and 4. Lake 3 appears more as though there was some partial freeze through of the sides of the lake. For lake 4, it is very hard to discern the lake in the Sentinel-1 image and makes me question whether there is indeed subsurface water here. What are the boundaries used for this lake?

Figure 4 – Do lakes 3 and 4 have enough backscatter data before the jump to indicate "sustained backscatter"?

Figure 5 – This analysis is extremely beneficial and I think it would be useful to show something similar for the other lakes in this study. Also, was the area used for each lake the area outlined in red in the NDWI Max Composite? This seems to miss what appears to be subsurface water for lakes C, G, and H. In fact, it seems that the subsurface part of Lake H also increases backscatter (although not as significantly as Lake 6).

L208 – "These reductions in maximum lake extent contrast with those observed for the many surrounding lakes, which fill to around the same size in adjacent summers". A

figure or some evidence of this would be useful.

L218 – What are the uncertainties on the elevation changes from photoclinometry? Do you have any idea why these values are so much larger than the depths from optical images?

Figure 6 – For Summer 2017 lakes 1 and 5: are these just cloudy images? If so, I would emphasize this somehow because it also looks like the lake just isn't there. Also, a scale would be nice. Once again, I do not find this analysis very convincing for lakes 3 and 4. You mention that they "change shape" but I do not see a significant shape change for lake 4.

Figure 7 – "elevation" should be added before "difference" in the first line of the caption

L269-271 – This is already mentioned and fits better in the methods section

L290 – can Sentinel-1 be used to determine if water is present in the lake at the start of the melt season? Of course it's harder to interpret than optical imagery but perhaps can give some idea of water presence?

L298 – Did you try DEM differencing? (https://doi.org/10.1029/2020GL087970)

L337 – "other hydrological phenomena" such as?

L343 – "what other types of behavior may indicate" is extremely vague

Figure B1 – Are the different colored dots significant? Also, please label the lakes in this image.

Technical corrections:

L26 – Needs a clarifier after 'This' to begin the sentence

L45 – "rising water levels in the lake" → "increased lake volume"

L58 – there is an extra space in "changes"

L93 – change "files" to "images"

L263 – "cover of cloud" → "cloud cover"

L324 – Sentence that begins with "This" with no clarifier

Figure A2 – Two periods at the end of caption

―――――――――――――――

---

## Author Comment (AC1) · 14 Aug 2020

**Responses to Reviewer 1**

**General comments**

This manuscript presents the first quantification of the drainage of supraglacial lakes in Greenland during winter. Such events have previously only been described qualitatively, or their occurrence inferred from proglacial river data. As such, the authors make a worthwhile contribution to help fill in some gaps in our understanding of ice sheet hydrology. The paper is on the whole clearly written and the data analysis is valid and suitable (barring a few inconsistencies – see specific comments below). The main conclusions are justified, although there are some overly speculative comments made at the very end of the manuscript.

**Thank you to the reviewer for his thorough and very helpful review of our manuscript and for these positive comments. We are pleased the reviewer recognises the 'worthwhile contribution' and 'filling in of gaps' our paper makes and are glad to hear he thinks it is generally 'clearly written' with 'valid' and 'suitable' data analysis with the main conclusions 'justified'. We will clear up the 'inconsistencies' and remove 'overly speculative comments' as detailed below.**

My main comment is that the temporal coverage of the radar data used is limited. Sentinel-1b only started consistently retrieving data from west Greenland in October 2016, so a 6-day period for the same relative orbit is only possible from then. This raises the question of why the authors did not look for winter lake drainages over more recent years (i.e. after 2016/17). Doing so might improve the temporal resolution of the data and thus avoid some of the limitations.

**The temporal coverage we include in our analysis spans 3 years. While, of course, time span can always be increased (as can spatial coverage) we note that many published papers investigating lake drainages or other phenomena on ice masses only cover 1 or 2 years. We are keen for The Cryosphere to publish what we believe is the first documentation of winter lake drainage on the GrIS. It will be up to others to adapt and extend this analysis to cover other time periods and other parts of the ice sheet, and other ice masses.**

**However, following the reviewer's comment we did investigate imagery from later years from the same relative orbit as we'd used in our analysis and unfortunately the temporal resolution is not significantly improved. We wish to examine just one relative orbit to remove ambiguity of backscatter associated with using different relative orbits. As a way of background, we first started this work in 2017 [Note Corinne Benedek has taken Maternity Leave since this time]. We chose an area of the ice sheet where others had worked and where we knew there were plenty of lake drainages. We chose a relative orbit where temporal resolution was good over the previous 3 winters, and that is how we arrived at the data set we have.**

**Specific comments**

L2: 'immediately' seems to contradict the 'hours to days' later in the sentence. I suggest removing it.

**We will replace 'immediately' with 'rapidly'.**

L3 & L26: Is meltwater access always sustained for the rest of the summer? If the ice is thick (so that creep closure rates at the base of the moulin are rapid) and surface meltwater input following lake drainage is low (i.e. the lake and moulin are at high elevation), the moulin might close and the lake refill.

**Recognizing this point, we will change line 3 to "and then can allow melt water. . ." and change line 26 to "may permit meltwater"**

L26: 'This' should be 'Drainage' otherwise it is somewhat vague what is being referred to.

**We will change "This" to "This drainage"**

L27: Not necessarily the 'down-glacier direction'. The direction of subglacial water flow is determined by the subglacial hydropotential surface, the slope and aspect of which will vary from that of the ice surface (due to the bed topography) and may be different from the broad definition of 'down-glacier'.

**We will change to "down-hydraulic-potential direction"**

L32: It might be worth adding that the ice speed often decelerates below the pre drainage value because of the temporary increases in basal hydraulic efficiency.

**We will add this suggestion**

L36: Although lakes contribute to total runoff from the ice sheet, they do not 'control' it. If you look at a seasonal hydrograph (e.g. Bartholomew et al. (2011, doi:10.1029/2011GL047063)), the overall shape is determined by atmospheric temperatures and ice surface melt rates. Because the highest melt rates are closer to the margins at lower elevations where there are fewer lakes, most meltwater enters the subglacial drainage system via crevasses and moulins not associated with lakes (Koziol et al. 2017). Lake drainages are typically superimposed on this seasonal pattern.

**We agree. Thank you. We will rewrite our text to make these exact points**

L48 - 49: This last part of the sentence doesn't quite make sense to me.

**Sorry this should read "Conventional understanding is that lakes that completely or partially drain during the summer then freeze during the winter, either freezing through completely or maintaining a liquid water core (Selmes et al., 2013; Koenig et al., 2015; Miles et al., 2017; Law et al., 2020).**

L52: You should use the final TC reference which is 2013 (also in the reference list).

**Yes, we will change this - The Cryosphere, 7, 1433–1445, 2013**

L64: More recent data acquisitions from Sentinel-1 a and b are more consistent and regular. Did you look over the 2017-2018 and later winters and not find any lakes? Or have you not looked at these data? Doing so might remove some of the temporal frequency limitations you mention later in the manuscript.

**Please see our response to this point in the General Comments section above**

L90: I wonder if it is worth mentioning somewhere that subglacial lake drainage (and the resulting formation of so-called 'collapse basins') might lead to a similar change in radar backscatter. The fact that you used a supraglacial lake mask to search for the backscatter changes suggests that the changes you identified were supraglacial lake drainages, but it might be worth a mention nonetheless.

**We think the place to make this point is not here in the Methods but perhaps in the conclusions / suggestions for future work and so we will add it there.**

L105. The 'therefore' does not quite follow as written, but needs more explanation in the previous sentence justifying why you'd expect gradual freezing to lead to an increase in backscatter. Also, you should provide more details about why you think that a lake drainage would lead to a sudden, significant and sustained increase in backscatter. Is it because the collapsed lid of the lake would create chaotic relief and therefore be bright, or is it just the change from the radar 'seeing' through the frozen to the lake surface, to the radar instead seeing the ice of the drained lake bed?

**We will remove the word "therefore". We will also explain more fully why we'd expect a slow lake freezethrough to be associated with a gradual backscatter increase. This is explained in the paper we were both involved with (Miles et al, 2017) but we will summarise things here and refer to that earlier paper. Briefly, liquid water absorbs HV backscatter, whereas frozen water reflects more of the signal** as bubbles entrained within frozen lake ice increase the relative backscatter compared to liquid water . The backscatter signal of unfrozen and frozen lakes is therefore sufficiently distinct to allow freeze-through identification.

Similarly, we will add a sentence or two with reference to previous literature about why **a lake drainage would lead to a sudden, significant and sustained increase in backscatter. We agree with the referee that both of his suggested processes are relevant. The former would produce a very high backscatter that is greater than the surrounding whereas the latter would produce an increase in backscatter to around the background values. We saw examples of both associated with summer lake drainages, which we reported in Miles et al 2017.**

L106: I think the comparison with a summer lake drainage is probably valid but requires a bit more explanation. In the summer case, the backscatter values change because the surface changes from water to ice. It is likely the same change that is seen in winter (even though the lake might be partially frozen over) because C-band SAR can penetrate a few m of ice - likely thicker than the frozen lake surface, at least in the early part of the winter.

**Yes we agree. We think this point is implicit in what we have said but we will make it more explicit.**

L121: It would be useful to also state the actual area in metres squared

**We agree and will add this, i.e. 8000 m². [Note the resolution of GRD scenes used is 40 x 40 m].**

L125: Should it not be the latest rather than the greatest? Otherwise the estimated volume might be significantly greater than it was at the time the lake drained. Later in the manuscript you do refer to the volume estimates being for the last Landsat image of the season, so I think there is a mistake somewhere here.

**Sorry there was an error made here in the description and in the calculation of area, depths and volumes. We agree that the calculations presented should be from the latest unfrozen Landsat-8 image prior to freeze over. We have recalculated areas, depths and volumes and will change the table values to those shown below. We will also add a listing in the appendix of the image scenes used for these calculations. Compared to previously the lake areas have all decreased. The exception is Lake 5, which has increased slightly, as a result of us accidentally excluding some peripheral pixels in the previous calculation that are now included. Compared to previously the mean lake depths have all increased and are now closer to the estimates derived from the photoclinometry method.**

| Lake | Location | Drainage Date | delta dB | z-score | Pre-drainage Lake Area | Pre-drainage Mean Lake Depth | Pre-drainage Lake Volume |
|---|---|---|---|---|---|---|---|
| Lake 1 | -47.32 , 68.70 | 11 Nov 2014 to 23 Nov 2014 | -4.3 | 3.5 | 0.04 km² | 0.57 m | 0.000021 km³ |
| Lake 2 | -48.52, 68.91 | 10 Jan 2015 to 22 Jan 2015 | -4.4 | 3.4 | 6.12 km² | 3.26 m | 0.0200 km³ |
| Lake 3 | -48.75, 69.43 | 05 Jan 2016 to 17 Jan 2016 | -3.8 | 2.7 | 0.43 km² | 1.89 m | 0.0008 km³ |
| Lake 4 | -48.38, 69.40 | 05 Jan 2016 to 17 Jan 2016 | -2.3 | 2.6 | 0.51 km² | 2.56 m | 0.0013 km³ |
| Lake 5 | -47.43, 68.62 | 10 Feb 2016 to 22 Feb 2016 | -3.2 | 2.8 | 1.84 km² | 0.86 m | 0.0016 km³ |
| Lake 6 | -48.03, 68.75 | 06 Nov 2016 to 18 Nov 2016 | -9.3 | 2.2 | 2.27 km² | 1.41 m | 0.0032 km³ |

L133: Did the image tiles include any seawater? If so, was this used as the darkest pixel? Might the darkest pixel not be from a lake with sediment at its base and thus not truly representative of the spectral signal of deep water?

**Yes, the tile included seawater; and yes, seawater was always the darkest pixel. We will amend the text to make this more explicit: "Reflectance of deep water was determined per image by selecting the darkest pixel (which was always a seawater pixel) in each image."**

L157 – 160: Understanding of this process would be greatly aided by the addition of an explanatory diagram.

We plan to tighten up the explanation of this method in the text. We mention that the method was used by Pope et al 2013 and described there (without ref to a diagram). But we can also add a simple 2D cartoon of a cross section along one of the transects shown in the Supp Mat Figure A2, first showing the offset and then showing closure of the offset and therefore the final surface . So Supp Mat Figure A2 would then have three components: a b and c.

L172 – 174: But you used the Landsat image with the greatest area for the lake depth rather than the latest one (L125). It is also possible that the lake volume reduced following your Landsat-derived volume calculation.

**Please see our response to the L125 comment above. Lake volumes have been recalculated based on the last available Landsat-8 image for each lake prior to freeze-over. The dates / filenames of these images will be included in the Appendix and referenced here as well. The text will be changed to reflect the volume calculation and to note that though they were based on the last available image before freeze over, this does not rule out the possibility that lake volume changed between the image acquisition date and freeze-over.**

L179: It would be useful to show the extent of the optical lake masks on the Sentinel-1 backscatter images to see over what area the mean change in dB is calculated. Also, might a median value be less prone to the influence of outliers?

**We agree that it would be useful to include the lake mask boundaries on Figure 3 and will add them. A draft edit of this figure is here and also included in the responses to Reviewer 2.**

[Figure]

Regarding using the mean vs median dB. We have checked the frequency distributions of dB for several lakes and they are normally distributed, with the mean and medians being the same value or only very slightly different. Please see distributions for the drained lakes below, showing the mean (dashed line) and the median (dotted line). We propose,

**therefore, to stick with our use of the mean and can justify this by summarising the above in the text.**

[Figure]

L186: 'identified' would be better than 'filtered out' (otherwise it seems like you are removing them from the time series)

**We will remove this sentence in response to the next comment**

L186 – 187: This repeats some of the methods section really. Is it needed here again? 'All other lakes. . .' could follow logically straight on from the previous paragraph.

**We will remove this sentence as the reviewer suggests**

Figure 3 caption: Does the last sentence definitely apply to this figure? It does not seem to make sense.

**Thank you for spotting this. The sentence does not belong here and will be removed.**

L198 (subtitle 3.2): It would be useful to state in the section title what you are confirming - 'Confirmation of winter lake drainage. . .'

**Thank you. We agree. The section title will be changed to 'Confirmation of winter lake drainage by optical imagery'.**

L219: Average depth for Lake 2 after drainage is more than double that calculated when the lake was present. Why do you think the differences are so large? Do you only calculate the depth of the depression to the lake shoreline using photoclinometry? Apologies if I've misunderstood the method, but I found it difficult to follow.

**Yes, using photoclinometry we only calculate the elevation change within the lake - so upto its shoreline. We will clarify this in the text. See also our reply to comment L157-160 above - we hope the method will be clearer with the addition of the extra diagram. Note also that the recalculations of the lake depths using the optical imagery with the last available image from the previous summer have resulted in slightly deeper mean lake depths (see our new Table 1 above) which bring them slightly closer to the mean depths calculated using the photoclinometry. However, it is still the case that the photoclinometry method of lake depth calculation produces lake depths that are bigger than those produced using the optical band method , > 2X for Lake 2, around 1.5 X for Lake 5 and nearly 2.5X for Lake 6.**

**We mention likely reasons for the discrepancies in lines 172-175 and also lines 294-303. These are all to do with errors in the two techniques of course. We propose to remove lines 172-175. Around L219 in the results we propose to say that possible reasons for the discrepancies will be discussed below in the Discussion. Then we will ensure that the errors in both the optical band method and the photoclinometry method and the likely reasons for the differences in lake depth calculations are discussed fully in the Discussion around what is now lines 294-303. We will quantify the depth errors in the two techniques with reference to previous literature. From Pope et al (2016) we estimate error using the optical band method is 0.46 m and from Pope et al (2012) we estimate error using the photoclinometry**

**method is 1.61. Please see our responses to Reviewer 2's comments for 'Table 1' and 'L 218' for derivation of these errors.**

**Finally, please note that these calculations of lake depth are subsidiary to the main point of the paper, which is to document winter lake drainages (rather than quantify precisely the volumes of water drained). These two additional 'tests' support the SAR backscatter changes by showing: i) that water depths were shallower in the subsequent summer than the previous summer; and ii) that surface elevation dropped over the winter.**

L227: 'calculated using' might be better than 'expressed through'

**Agreed. We will change the text as suggested.**

Figure 7 caption: The second and third sentences are a bit convoluted. I suggest changing to: 'The first column of images shows the collapse vertical distance of each pixel calculated by interpolating and differencing the pre- and post-drainage topography.'

**Thank you. We agree and will alter the text as suggested.**

L232: I think it would be worth briefly reiterating how you used the z-score – i.e. the z-score of backscatter change for each lake is calculated relative to the backscatter change of all lakes across the scene

**This is a good idea and we will reiterate briefly what we mean here and how we used the z-score to identify large, anomalous and sudden changes. The point here, of course, is that we also need to ensure the changes are also sustained to identify lake drainages correctly.**

L250: C-band SAR penetrates a few m of ice (Rignot et al. 2001), so likely sees through the nascent ice lid. I think this needs to be stated more clearly early on. You discuss the low backscatter values in a somewhat vague manner initially before offering an explanation in Section 3.3.3. Perhaps it would make more sense to swap the order of Sections 3.3.2 and 3.3.3?

**We would like to keep the order of sections 3.3.2 and 3.3.3 as this is the order in which the methods are done and the images are processed (equivalent to methods sections 2.1 and 2.2). and the method proceeds.**

**We will add a bit more to the end of the methods section (2.2), when we talk about using HV polarisation data to image shallow subsurface lakes, that HV data penetrates several metres through the surface, including snow, firn and any nascent lake ice lid. We will add the Rignot et al reference there. We will also add to the sentence on L250 to reiterate that we're using HV polarisation data, which is sensitive to volume scattering and therefore may be detecting water below the surface not seen in optical imagery. See also our response to comment L105 and L106, where we propose to clarify that HV backscatter changes are due to shallow subsurface processes.**

L257 – 258: Based on Figure 5 you might have more luck using Otsu thresholding on the Sentinel-1 images, as this would 'fill in' the interior of many of the lakes that are doughnut shaped in the NDWI composite.

**We thought of this but decided not to make the assumption that doughnut-shaped or other irregularly-shaped lakes necessarily contained water beneath a snow/ice lid. We wanted to focus the analysis of backscatter change solely on those areas which irrefutably showed evidence for deep water in the optical images. Using the Otsu thresholding method to 'fill in' lake interiors would have dampened the backscatter change signals we found if, in fact, those areas were not actually part of a lake. We would then have had 'less luck' in finding lake drainage events.**

L266: The value of 9 m is for dry cold firn. It will be less for the ice lids on the lakes (a few m or less I expect based on Rignot 2001).

**Yes we will change the text accordingly and refer to "a few metres"**

L272: Be clear that this is temporal frequency

**Thank you, yes, we will add the word "temporal" to refer to "temporal frequency" here.**

L273: Both satellites were only recording image consistently from c. October 2016

**Agreed.**

L282 – 283: Here you state that the depth estimates were based on the last available image, but on L125 you state that the depth measurement was based on the image when the lake was largest.

**As per the earlier comments, we have corrected the area, depth, and volume calculations presented in Table 1 to show quantities based on the last available Landsat-8 image before freeze-over.**

L285 – 287: Based on the above discrepancy in how you measured the lake depth, your estimate might very well be an overestimate rather than an underestimate. This needs to be cleared up and the justification of why the lake depth and the photoclinometry depth are so different amended accordingly.

**Please see our response to comment for L219. We will add error estimates to our calculations of water depths based on both the optical band and the photoclinometry methods. We will clarify why the optical band method may underestimate water depths (crucially there is a depth threshold beyond which light attenuation is unaltered - Pope et al 2016; Williamson et al, 2018) and why the photoclinometry method may overestimate water depths (differences in the date of the DEM and the dates of the imagery used to calculate the slope-reflectance relationships; and shadowing in the lake basin not seen**

**outside of the lake basin introducing error in slope calculations inside the lake basin when using an empirical relationship defined for areas outside the lake basin).**

L290 – 291: In terms of determining whether water was transported into the basin from higher elevations, could you not compare the dB values with the maximum achieved over the winter to detect surface melt at higher elevations? You could also use the runoff output of a regional climate model like RACMO.

**Both of these things could be done but we think they are not relevant to and would therefore detract from the main purpose of our paper. The main point of our paper is to provide what we believe to be the first method for identifying automatically lake drainages using changes in SAR backscatter within lake basins. This has not previously been reported in the literature. Furthermore, we have applied the method and identified winter lake draianges. This phenomenon has not previously been reported in the literature either. We wanted to verify our method using other remote sensing techniques, which we have done using available optical imagery in two different ways. First, we have shown that water depths in the lakes prior to winter drainage in the previous fall are greater than those after drainage in the subsequent spring. Second we have used photoclinometry to show that there is a collapse in the lake surface elevation over the winter. In response to a comment by both referees, we have also used 2m resolution ArcticDEM strips to verify elevation change associated with the drainage of Lake 6 (see below). So providing a new method, applying it, and verifying it is the purpose of our paper.**

**The calculations of water depth and volume are very much a subsidiary part of the paper, but we provide these for general interest.**

**We think the referee is implying that dB values of SAR imagery (presumably imagery collected at the same time as the first available optical imagery the following spring) could be used to determine whether there's been any lake filling between the time of the winter lake drainage and the time of the 1st available optical image in the spring. This could be done but it would still not allow us to adjust the optically derived lake depth to allow us to get a better estimate of drained lake volume. The same procedure would have to be applied between the last available optical satellite image the previous autumn/fall, and the time of the lake drainage to determine whether water entered the lake (or froze in the lake) over the intervening period. Again, we would not be able to quantify the volume of water involved.**

**The referee also talks about using RACMO to adjust the lake volumes determined from optical imagery. What we assume he's thinking about here is that runoff into the lake basin between the time of the lake drainage and the time of the first available optical image could be used to adjust the lake volume derived from the optical image according to how much extra water may have flowed into the lake during the spring. Again, presumably the same would need to be done between the time of the last available optical image the previous autumn/fall and the time of the lake drainage in order to adjust the lake volume derived from the autumn/fall optical image according to how much extra water may have flowed**

into the lake. What would be required here, is actually a surface hydrology routing model driven by the runoff output from RACMO. This, we believe, is way beyond the scope of this paper. All we are trying to do in the final section of our paper is use independent evidence to verify that a winter lake drainage occurred. We could leave it there but we thought it would be useful to obtain first order approximate values for the volume of the lake drainage event, which we do. And we discuss the errors associated with the derived volumes.

L293 – 294: Have you considered using the ArcticDEM time-stamped data strips? There may be some that would help to further constrain the volume of the drained lakes. See e.g. Livingstone et al. (2019) https://doi.org/10.5194/tc-13-2789-2019

Thank you for this suggestion. We have examined the ArcticDEM time-stamped 2 m data strips and a satisfactory pair of 'before' and 'after' images exists only for Lake 6. In this case, a marked difference is shown in Lake 6 surface elevation before and after drainage. Using ArcticDEM 2 m strips from 21 September 2016 (before drainage) and 12 March 2017 (after drainage), we calculate the elevation difference (after minus before) seen in the figure below. If we mask this by the lake mask for Lake 6, we get a mean before/after depth difference of 2.17 m. Note this compares with the mean depth derived from the optical band method of 1.41 m (new Table 1 - see above) and that from the photoclinometry method of 3.38 m (Fig 7 and stated on L219). Note also that this photoclinometry-derived value is less than that quoted in our original manuscript (4.04 m) where we had not masked the lake according to our optically-derived maximum composite lake mask. For Lakes 2 and 5, the optically-derived lake masks are the same as those over which we apply the photoclinometry method. For Lake 6, the optically-derived lake mask is smaller than than over which we apply the photoclinometry. To compare with the optically-derived mean depth estimate, we must crop the photoclinometry-derived and the ArcticDEM-derived depth estimates. We will adjust our manuscript in the relevant places to explain this and make the correct comparisons.

The Figure below is a draft. We propose to add a figure to our paper for Lake 6 which is similar to the current Fig 7. So it will have 3 panels, elevation change and hillshades of the before and after ArcticDEMs for lake 6 and surrounding area. We will adjust the colour bar to be the same as that in Fig 7.

[Figure]

Lake 6

-10 m          + 10 m

0   2   4 km

L301: Maybe remind the reader that this refers to the 5 m mosaicked product so is made up of data from many different times.

**Will will change the line to read ". . . in the ArcticDEM 5 m mosaics."**

L303: Changes in backscatter are 'caused by' lake drainage events

**Agreed.  We will add 'caused by' lake drainage events.**

L313: What about short sharp melt events over winter? Have you looked at any available meteorological data? Also do you detect a reduction in backscatter for the non lake surface at the same time the lake backscatter increases? This might indicate a small amount of surface melting that might have an effect on the (presumably relatively inefficient) subglacial drainage system if it got to the ice bed.

**We had not looked at meteorological data to see if there's evidence for short melt events coinciding with lake drainage events during the winter. But following the referee's suggestion, we have examined the Swiss Camp air temperature record for the 6 month (Oct-March) periods 2014/15, 2015/16, and 2016/17 covering our 3 winters (see the Figure below, where the 12 day periods during which lakes drain are indicated by the vertical lines).**

**As you can see, there is no clear evidence that the 6 lake drainage events are associated with especially large increases in air temperatures to above zero that would be indicative of melt events. The only exception might be Lake 1 where there is a large rise in temperature**

from -21 to near zero and the highest temperatures since early-mid Oct. But for the rest, air temperatures are either rising but not to above freezing, falling, or fluctuating. Furthermore there are other larger rises in air temperature sometimes rising to zero at other times of the year that are not associated with lake drainage events.

**Given this, we do not propose to include this analysis in our paper, but we can in Supp. Mat. if the referee or editor thinks it would be helpful.**

[Figure]

L316: The transient nature of any speed-up probably means that there would be no discernible signal in a winter average velocity estimate.

**We think the referee has misunderstood what we're trying to say here, which we agree is not very well articulated. We are not suggesting that we might see the effect of a single lake drainage triggered speed up in the MEaSUREs data set. We are using the MEaSUREs velocity field to see if the locations of lakes are in particularly fast flowing areas of the ice**

**sheet or areas of high strain rates which they do not seem to be. We propose to rewrite this section to make it clearer.**

L317 – 318: I'm not sure your sample size is big enough to be able to say this definitively, so it may be worth including this caveat.

**We will change the text to read "No pattern of lake locations and speeds seems to be visible, although our sample size is small and more evidence is needed to examine this possible association further".**

L322 – 325: Without actually doing a rough calculation (basin or lake diameter, velocity and time) this seems overly speculative.

**We agree, and given the referees previous comment about small sample size we do not think it worth performing these calculations and so we propose to delete these sentences.**

L319: the term 'cascade draining' is a little misleading (although I realise it is used in the title of the Christoffersen paper). Perhaps add a very brief explanation of the process – i.e. drainage of one lake creates ice acceleration and a tensile shock that is transferred through the ice and can trigger other lakes to drain etc.

**We will change the text to: "These concurrent drainages support the observations and modelling of Christoffersen et al. (2018) where the drainage of one lake creates localised ice acceleration, which is transferred via stress gradients to other areas triggering other lakes to drain". Alternatively, they may indicate a larger scale ice movement that triggered both events simultaneously."**

L329: I don't think it is necessary to repeat 'large, sudden, anomalous and sustained' here.

**Agreed. We will change the text to: " We find six winter lake drainage events across a study site containing approximately 300 supraglacial lakes".**

L329 & 332: I think it is worth specifying that you are talking about supraglacial lakes here (for anyone who might just read the conclusion).

**Agreed. We will add 'supraglacial' after '300'.**

Technical corrections: Figure 2: Lines need to be thicker and symbols larger (and C is very difficult to see)

**Agreed. We will make these changes to make the graph clearer.**

L148: missing space between value and units

**Agreed. And we will check the entire document for this.**

L151: Do you mean Appendix A? Appendix B appears to show ice velocity data.

**Agreed, we will correct this to read Appendix A3**

L230: 'event' should be 'events'

**Agreed. We will edit the text to 'events'.**

L242: 'false negative ones' should be 'false negatives'

**Agreed. We will edit the text to 'false negatives'.**

L243: 'false positive' should be 'false positives'

**Agreed. We will edit the text to 'false positives'.**

---

## Author Comment (AC2) · 14 Aug 2020

Responses to Reviewer 2

General comments

This manuscript presents evidence of 6 different winter lake drainages across the Greenland Ice Sheet. The authors use a variety of methods (Sentinel-1 backscatter tracking, optical imagery analysis, photoclinometry) to provide evidence of these lake drainages and quantify drainage volumes. The findings presented in this paper are a valuable contribution to better understanding Greenland Ice Sheet hydrology. My first concern with the paper is that the writing is, at times, hard to follow. This is particularly true within the Methods section where overly wordy sentences take too long to dissect and comprehend. My second concern is that I am not convinced by the evidence for paper the 'drainages' of lakes 3 and 4 for reasons which I have further discussed below. Additionally, there is no elevation-change analysis from photoclinometry for these lakes. I understand that this may not be possible with the available Landsat-8 images; however, I don't believe that the evidence presented is convincing.

**Thank you to the reviewer for spending the time so carefully looking through our manuscript. We're pleased the reviewer thinks our paper makes a 'valuable contribution' to the understanding of GrIS hydrology.**

**Regarding the writing. We propose to go through the manuscript very carefully clarifying all places where this referee and the other referee did not immediately grasp what we had done. We could also add a 'flow chart' type Figure to our Methods section if the reviewer/ editor felt this would be useful.**

**Regarding Lakes 3 and 4. We agree that the evidence for winter drainage of these lakes is more equivocal than for the other 4 lakes. However, the backscatter change for these lakes does meet what we think are quite strict rules for defining lake drainage, i.e. a large, anomalous, sudden and sustained increase in backscatter. One of our rules is that there should not have been a big reduction in backscatter immediately prior to a large increase. We look only at the time interval immediately prior to the increase to detect whether or not there has been a large previous decrease. This runs the risk of error of commission, which is *possibly* the case for Lakes 3 and 4. However, as Figure 4 shows, Lake 1 is also picked up as a draining lake using this criterion, and this lake drainage is then very well supported by the additional evidence. We *could* redefine our definition of a lake drainage to say that we need evidence from *two* prior time steps rather than just one. This would then have excluded Lake 1 from our analysis as there is only one image prior to the large jump in backscatter to look at (as there is for Lakes 3 and 4). So with this stricter definition we would have errors of omission. Given all this, we would like to propose that we keep our current definition of a lake drainage so that we can include Lake 1. This will mean we also keep and show Lakes 3 and 4. *But* we will then in the discussion and conclusions highlight even more forcefully that the extra optical band evidence in support of winter lake drainage is lacking (although lack of evidence isn't necessarily proof of course). We will articulate what we say above regarding the criteria for identifying lake drainage and errors of omission / commission and change text accordingly. For example, we will reorientate**

**L339-342 to suggest we may have included false positives (Lakes 3 & 4) but a stricter requirement regarding prior imagery might include false negatives (e.g. Lake 1). If people wish to use our technique in the future they can decide whether to be more or less strict by scrutinising either 2 or 1 images prior respectively. We hope that the referee / editor agree this is a good way forward.**

Specific comments

L7 – specify which winters

**We will change the text to the following: ". . . during the three winters (2014/15, 2015/16 and 2016/17) in fast flowing parts . . ."**

L36 – Sentence beginning with "Lake drainage events, therefore,. . ." seems out of place within the rest of this paragraph.

**Reviewer 1 also commented on this sentence. We will change the sentence to "Thus, lake drainage events influence the quantity and quality of water issuing from the ice sheet, although their effects are superimposed on the larger scale atmospheric controls on melt patterns and runoff".**

L37 – Where do drainage events raise levels of phosphorus, nitrogen and sulfate?

**We will edit this to read as follows: ". . . raise levels of phosphorus, nitrogen and sulphate in proglacial streams (Hawkings et al., 2016, Wadham et al., 2016), . . ."**

L43 – I don't believe Koenig et al (2015) documented lake drainages, just the existence of winter-stored meltwater.

**We will remove the reference to Koenig et al. 2015 here and keep it in line 49 in discussion of winter lake freeze-through.**

L44 – Perhaps combine these two sentences so the second one doesn't start with "They".

**Agreed. We will combine the sentences.**

L47-49 – The sentence beginning with "conventional understanding" does not make sense

**Apologies, this was a typographical error and was spotted by Reviewer 1 too. The sentence should read "Conventional understanding is that lakes that completely or partially drain during the summer then freeze during the winter, either freezing through completely or maintaining a liquid water core (Selmes et al., 2013; Koenig et al., 2015; Miles et al., 2017; Law et al., 2020).**

L51-53 – This sentence is somewhat unclear to me

**We will split the sentence and change the text to "Proglacial stream evidence from one study suggested that water was released from englacial or subglacial stores (Rennermalm et al., 2012). Proglacial stream evidence together with the appearance of surface collapse features on the ice sheet suggested that water may have been released from surface lakes (Russell, 1993)."**

L54 - delete "carefully" and "in" in "microwave backscatter in Sentinel-1 satellite"

**Agreed. We will remove these words.**

L76 – what are the dates that determine a "late season" image?

**We will include the dates of the images used within the Appendix. Late season images ranged from ~ July 25 through August. We began with images from the last week of August alone and added earlier images as necessary to achieve cloud-free coverage of the full site.**

L86 – Would it make more sense to use the last optical image from the summer to define the lake boundaries instead of the maximum?

**We think it best to use the composite image rather than just a single late summer image to define the lake areas within which to then look for SAR backscatter change. It means the lake areas are defined on the basis of a few images rather than just one, which will remove possible errors associated with relying on just one image. The date of the last image may vary due to variable cloud cover. Using just the last image does not allow for the possibility that the lake fills after the last available image. It also means we're looking at the mean dB change over a larger area and so we'll be erring on the side of caution when defining a dB change. It also allows for the possibility that the last image extent may underestimate the true water extent that can be detected in the SAR imagery if water around the lake edge is shallow subsurface and not visible in the optical image.**

L111 – What does "lakes across the scene" mean? How large of an area is this?

**For clarity, we propose to change this to " . . . all lakes within the study site . . .". We describe the size of the site and the number of lakes earlier on lines 61-62.**

L112 – What does the last sentence mean?

**Apologies for the confusion. We propose to delete this last sentence as the relevant points of the method are addressed in line 120 in the following paragraph.**

L125-126 – Again, would it make more sense to use the last optical image from the summer to calculate lake volumes instead of the maximum lake area?

**Sorry - the areas, volumes, and depths shown in Table 1 contained an error in the submitted manuscript and along with the error a mistaken description of the images used for calculating depth. We agree that for this calculation the last available image prior to freeze-over is the most appropriate as it most closely represents the volume of water present in the lake at the time of drainage. We will be editing Table 1 and the description of images used here and in the Table caption to reflect this correction. Table 1 values will be changed to appear as follows:**

| Lake | Location | Drainage Date | delta dB | z-score | Pre-drainage Lake Area | Pre-drainage Mean Lake Depth | Pre-drainage Lake Volume |
|---|---|---|---|---|---|---|---|
| Lake 1 | -47.32 , 68.70 | 11 Nov 2014 to 23 Nov 2014 | -4.3 | 3.5 | 0.04 km$^2$ | 0.57 m | 0.000021 km$^3$ |
| Lake 2 | -48.52, 68.91 | 10 Jan 2015 to 22 Jan 2015 | -4.4 | 3.4 | 6.12 km$^2$ | 3.26 m | 0.0200 km$^3$ |
| Lake 3 | -48.75, 69.43 | 05 Jan 2016 to 17 Jan 2016 | -3.8 | 2.7 | 0.43 km$^2$ | 1.89 m | 0.0008 km$^3$ |
| Lake 4 | -48.38, 69.40 | 05 Jan 2016 to 17 Jan 2016 | -2.3 | 2.6 | 0.51 km$^2$ | 2.56 m | 0.0013 km$^3$ |
| Lake 5 | -47.43, 68.62 | 10 Feb 2016 to 22 Feb 2016 | -3.2 | 2.8 | 1.84 km$^2$ | 0.86 m | 0.0016 km$^3$ |
| Lake 6 | -48.03, 68.75 | 06 Nov 2016 to 18 Nov 2016 | -9.3 | 2.2 | 2.27 km$^2$ | 1.41 m | 0.0032 km$^3$ |

L175 – I imagine that partial re-freeze would greatly impact the lake volume. Some water must have frozen as these lakes are no longer on the surface but are buried beneath a layer of ice. Also, I am wondering how the lake area detected from optical imagery compares with lake area detected from S1 imagery immediately prior to collapse? I imagine that the outlines of lake 3 and 4 would look quite different between the optical and S1 imagery.

**Regarding refreezing. We agree that a partial refreeze between the time of the last available satellite image in the previous summer and the time of the lake drainage in the winter would impact the lake volume. The depth of refreezing cannot be gleaned from satellite imagery. A model would be needed to calculate this. However the focus of our paper is not on the precise volume of water drained, but on the fact that winter lake drainages occur at all. Here we are using the optical imagery in the way we do to get simple estimates of the drained lake volumes, which we can then compare with the other estimates of drained lake volumes from photoclinometry. It is encouraging that both methods give not dissimilar results showing that L2 > L6 > L5 in terms of volume drained. We note, however, that the optical band method *underestimates* lake volumes compared to the photoclinometry method so the role of refreezing is likely less important than the fact that the optical method is biased towards measuring shallower water depths due to possible under-measurement of the deepest water because of saturation of the red band within the water column (Moussavi et al., 2016; Pope et al., 2016). As we say in reply to a comment on this section by referee 1, we're proposing to remove lines 172-5 here and discuss the reasons for the differences between the two volume estimates more fully in the Discussion.**

**Regarding lake area. It does appear from the images that there is a difference in lake outlines between the optical and the SAR data. Outlining the precise boundary of a supraglacial lake based on SAR imagery alone is not straightforward, and at the present time there is no published method for delineating the lake outlines from SAR imagery alone. This is the subject of ongoing work. For this work, we bound the lakes using optical imagery in line with established published methods and used these outlines to track SAR backscatter changes over time (e.g. Miles et al, 2017). It seems from the imagery that water exists under the surface where it is not evident in the optical data, but we cannot be certain this is the case. Further work is needed to establish methods to determine water presence in subsurface lakes where none is visible in optical imagery.**

**For clarity (and in response to a comment from the other Reviewer as well) we intend to add the optically-determined lake mask onto Figure 3 to better illustrate the area of analysis.**

Table 1 – What are the uncertainties on lake depth and volume?

**For the optical band method shown here in Table 1, we will use the values from the detailed error assessments undertaken for the Greenland Ice Sheet by Pope et al, 2016.**

**https://tc.copernicus.org/articles/10/15/2016/**

**They calculated errors for Landsat 8 data of 0.28 m for the red band and 0.63 m for the panchromatic band. As we're using the averages of the red and panchromatic band in our work (as recommended by Pope et al, 2016) we will assume an error of (0.28 + 0.63) / 2 = 0.46 m.**

**We will add these errors to the depth calculations shown in Table 1 and use them to estimate errors for our calculations of lake volumes. In line with previous work using these methods, we do not define errors for lake areas, which instead are fixed according to our threshold NDWI$_{ice}$ value of 0.25.**

L190 – With regards to Lake 6: I looked briefly at this lake on GEE during this time period using the HH band. I noticed that surrounding lakes show an increase in backscatter similar to lake 6 with the HH band. Do you have an explanation for this?

**HV polarised SAR accentuates volume (shallow subsurface) scattering whereas HH polarised SAR accentuates surface scattering. So an increase in HH backscatter of all lakes probably reflects an overall increase in surface roughness (formation of sastrugi for example) whereas the increase in HV backscatter picks out the reduction in volume scattering due to the drainage of water.**

**https://nsidc.org/sites/nsidc.org/files/files/SARTheory.pdf**

Figure 3 – I believe it would be useful to include dates on these images. Also the last line of the caption seems misplaced. Finally, I am not convinced by the 'drainages' of lakes 3 and 4. Lake 3 appears more as though there was some partial freeze through of the sides of the lake. For lake 4, it is very hard to discern the lake in the Sentinel-1 image and makes me question whether there is indeed subsurface water here. What are the boundaries used for this lake?

**We will edit the figure to include dates. We will also add the lake boundaries (this suggestion was also made by Reviewer 1). The figure below shows the proposed changes. Please see our detailed response to the general comment at the start of this review above regarding the issue of whether to include Lakes 3 and 4. They are highlighted by our analysis as having large, anomalous, sudden and sustained changes in backscatter, that are unlike those observed in other lakes. We propose to keep them in our paper, but be more circumspect with regards to their interpretation. Including them as "possible" lake drainages may help others who may wish to use / adapt our technique for use in other years and / or other areas of the ice sheet.**

[Figure]

Figure 4 – Do lakes 3 and 4 have enough backscatter data before the jump to indicate "sustained backscatter"?

**This is a good question. Please see our detailed response to the general comment at the start of this review.**

Figure 5 – This analysis is extremely beneficial and I think it would be useful to show something similar for the other lakes in this study. Also, was the area used for each lake the area outlined in red in the NDWI Max Composite? This seems to miss what appears to be subsurface water for lakes C, G, and H. In fact, it seems that the subsurface part of Lake H also increases backscatter (although not as significantly as Lake 6).

**We agree that these figures would be useful and will plan to include them in the supplementary material for the other lakes.**

**Yes, the area used for each lake is the area outlined in red in the NDWI Max Composite. We agree that it is possible we are missing some areas of subsurface water but at the moment there is no published method for identifying whether a pixel contains subsurface water or not from Sentinel-1 imagery alone. From the optical imagery, it is not possible for us to know for certain whether 'non-water' areas are floating ice covered by snow, or genuine ice islands or peninsulas. For this reason, we opted to confine our analysis to deep water demonstrated by optical data.**

L208 – "These reductions in maximum lake extent contrast with those observed for the many surrounding lakes, which fill to around the same size in adjacent summers". A figure or some evidence of this would be useful.

**We will plan to include a supplementary graphic which is similar to Figure 6 but with an altered scale to include more surrounding lakes. See below for a draft of such a Figure - we will add arrows or a box to highlight the drained lakes. In this additional figure, the background image in each is the maximum NDWI composite for the given summer season and the red shaded region is the lake mask used for analysis based on an NDWIice value in the summer composite >= 0.25.**

[Figure]

L218 – What are the uncertainties on the elevation changes from photoclinometry? Do you have any idea why these values are so much larger than the depths from optical images?

**For the photoclinometry method we will use uncertainty values from the detailed error assessment undertaken for Langjökull, Iceland by Pope et al (2012, their Table 2)**

**https://www.tandfonline.com/doi/pdf/10.1080/01431161.2012.705446**

**Here they compared elevations derived using the photoclinometry method on Landsat imagery, with airborne LiDAR elevation data. In areas where the photoclinometry assumptions were met (no shading) the median error is just 0.03 m, so the height difference error is then sqrt(0.03^2 + 0.03^2) = 0.04 m. In areas where photoclinometry assumptions were not always met (e.g. shaded areas), the median error is 1.44 and the equivalent height difference error is 1.61 m. We suspect the real error for our case on the Greenland Ice Sheet lies somewhere between these two, but to account for the different locations, DEMs, solar elevations and along-track spacing of the tie points between the Iceland and Greenland studies we will use the larger of the two errors, i.e. 1.61 m. We will add these errors to our calculations of lake depths and also use these to estimate errors for our calculations of lake volumes. In line with previous work using these methods, we do not define errors for lake areas, which instead are fixed according to our threshold NDWI$_{ice}$ value of 0.25.**

**We do have ideas about why these values are larger than those derived from the optical band method which we discuss on lines 294-303. We plan to improve the explanation for the possible reasons for the discrepancy here - please see our response to Reviewer 1's comment for L219 and photoclinometry uncertainty.**

Figure 6 – For Summer 2017 lakes 1 and 5: are these just cloudy images? If so, I would emphasize this somehow because it also looks like the lake just isn't there. Also, a scale would be nice. Once again, I do not find this analysis very convincing for lakes 3 and 4. You mention that they "change shape" but I do not see a significant shape change for lake 4.

**We will add a scale to this figure. We don't know for sure whether the 2017 images for lakes 1 and 5 are cloudy or whether the lakes are largely obscured due to snow blowing and drifting or to recently settled snow. Note these lakes are in the same general area of the ice sheet and at the highest elevations of the 6 lakes. We will add a note to the figure heading to point out the possible reasons why the lakes may be obscured.**

Figure 7 – "elevation" should be added before "difference" in the first line of the caption

**Agreed. We will add "elevation" in the caption.**

L269-271 – This is already mentioned and fits better in the methods section

**We assume the reviewer is referring to:**

**"Occasionally, images showed large scene-wide departures from typical backscatter values. These images (dated: 03 Feb 2015, 10 Apr 2016, and 16 May 2016) were omitted in this**

study as they were anomalous although if it were known what caused this phenomenon then perhaps the images could be corrected and used."

It is the last part of this sentence that is part of the discussion here and which we'd like to state as it is a 'problem' that needs to be overcome. We suggest shortening the sentence to:

"In our study, three images showed large, scene-wide departures from typical backscatter values and were omitted from further analysis. If it were known what caused this phenomenon then perhaps the images could be corrected and used."

L290 – can Sentinel-1 be used to determine if water is present in the lake at the start of the melt season? Of course it's harder to interpret than optical imagery but perhaps can give some idea of water presence?

There is no published method for determining whether a given pixel contains water from Sentinel-1 backscatter values alone. While work is being done to address this question using additional data and/or machine learning, it is not a trivial issue. For the purposes of this study, we decided to confine our work to pixels that we can verify as water through optical data. We could speculate about the behaviour of the water based on what we can see in the backscatter patterns, but without sufficient evidence we are reluctant to do that.

L298 – Did you try DEM differencing? (https://doi.org/10.1029/2020GL087970)

Reviewer 1 (his comment for L293 – 294) also suggested we try DEM differencing using individual ARCTIC DEM 2 m strips. Please see our detailed response to his comment. We were able to find before and after lake drainage strips only for Lake 6. We have performed the DEM differencing for Lake 6 and the results confirm a mean lowering of 2.17 m , adding further weight to our algorithm for detecting lake drainages from SAR imagery.

L337 – "other hydrological phenomena" such as?

we will add "such as onset of melt, rapid filling, or rate of freezing"

L343 – "what other types of behavior may indicate" is extremely vague

We will delete this sentence.

Figure B1 – Are the different colored dots significant? Also, please label the lakes in this image.

We will edit this image to replace the colored dots with lake numbers.

Technical corrections:

L26 – Needs a clarifier after 'This' to begin the sentence

**Will change to "This lake drainage and subsequent water input generates…"**

L45 – "rising water levels in the lake" → "increased lake volume"

**Will make this change.**

L58 – there is an extra space in "changes"

**Will remove space.**

L93 – change "files" to "images"

**Will make the change.**

L263 – "cover of cloud" → "cloud cover"

**Will change.**

L324 – Sentence that begins with "This" with no clarifier

**Will edit to say "This finding..."**

Figure A2 – Two periods at the end of caption

**Will edit.**

---

## Referee Report (RR1)

I thank the authors for accommodating some of my and the other reviewer's suggestions in this revised version of the manuscript and I think the manuscript is much improved. I only have a few minor comments that I would like the authors to address:

L 13: change "demonstrating" to "which demonstrates"

L 55: I suggest adding a reference to Lampkin et al. 2020 here (https://www.frontiersin.org/articles/10.3389/feart.2020.00370/full)

L 57 and 60: change "suggested" to "suggests"

L 122: I suggest adding a reference to Dunmire et al. 2020 here (https://agupubs.onlinelibrary.wiley.com/doi/10.1029/2020GL087970)

L 138: Sentence that begins with "Lakes were also checked" is confusing

L 161: Section 2.5: I suggest renaming this section "Elevation change from photoclinometry". When I first read the section header I thought that both photoclinometry and DEM differencing would be included in this section. I think the "and elevation change" is somewhat misleading.

L 162: Change "This technique: to "photoclinometry:

L 236: Change "signifying" to "signify"

Figure 3: In the caption, please comment that the red outlines signify the lake margins detected from optical imagery.

L 266: Section 3.3: I suggest changing "the ArcticDEM" to "ArcticDEM differencing" in the section heading

L 281: The sentence beginning with " The use of an NDWI_ice – based mask…" seems out of place in this paragraph and I got lost while reading this part.

Figure 7: I suggest using a non-diverging color scale when indicating elevation changes between 0 and 12.

Figure 8: It is more appropriate to use a diverging color map that is symmetric (centers on 0). At first glance it appears that the green indicates an increase in elevation everywhere outside the lake but I believe this is not the case?

L 229: Change "changes in SAR backscatter" to "SAR backscatter changes".

L 469: Appendix section A4: What do you mean by "Process" for this section header?

Appendix Figure B1: Again, I suggest using a non-divergent color scale to indicate velocity between 0 and 1000 m/yr.

Appendix Figure C1: Comment that the red boxes surround the lakes in this study.

---

## Author Response (AR2)

Referee # 1 Comments

L10 (and elsewhere): I wonder if the lake volumes would be better quoted in metres cubed? The cubic kilometres numbers are very small indeed, and on line 27 discharge rate from lakes is discussed in cubic metres per second.
CB response: change made as above

L14-15: I'm not convinced that the paper does show that 'background winter ice motion can trigger rapid lake drainage'. Rather the paper indicates that lake drainage occurred in the absence of surface melting and discernible ice flow acceleration.
CB response: Line has been edited to now read "The findings show that lake drainage can occur in the winter season in the absence of active surface melt and notable ice flow acceleration. . ."

L19: 'by reducing' would be more specific than 'via their effects on'.
CB response: change made as above

L53: Missing space.
CB response: change made as above

L54-55: I suggest that '. In another study, proglacial stream evidence from one study suggested that water was released from englacial or subglacial stores' is deleted.
CB response: Have deleted and moved Rennermalm reference to previous sentence, as appropriate.

L56-57: The wording here suggests that this evidence was linked to the Rennermalm study, which I assume (based on the dates of the studies) it was not.
CB response: edited to read "On another occasion, proglacial stream evidence together with the appearance of surface collapse features on the ice sheet were used to suggest that . . ."

L102: 'Orbit File Application' (to fit with the way the other stages are worded).
CB response: We are choosing to keep the phrase in this form as this is the precise wording in the SNAP workflow that is used by Google Earth Engine.

L103: 'Geocoding' or 'Orthorectification' might be a better term as otherwise this could be confused with correcting backscatter for local terrain-derived incidence angle.
CB response: Added 'orthorectification' in parentheses for clarification but retained original Terrain Correction as that is the name of the step in the SNAP workflow used by GEE.

L114-115: This sentence is slightly repetitive and awkwardly worded.
CB response: edited to read "To examine changes in lake behaviour, we created a time series of mean backscatter for each lake through each winter season using Sentinel-1 imagery."

L115: 'over the winter' might be better after 'process'.
CB response: Changed as suggested

L152: 'summed' might be better than 'added'.
CB response: agreed. Changed as above

L187: This error seems to have been calculated by adding the component errors in quadrature (which is the correct method). The same method should be used for calculating the error in differencing the ArcticDEM (see comment below).
CB response: Agreed. Change has been made as above.

L192: ICE should be subscript.

CB response: Changed as above.

L199: Should the DEM errors not be added in quadrature to determine the error in the DEM difference calculation? sqrt(0.2.^2 + 0.2.^2) = 0.28 m
CB response: Agreed, change has been made as above.

L227: 'single image transition' - state what time period this is - 12 days?
CB response: Changed as above

L264: 0.08 should be 0.28 I believe if the correct error calculation is used.
CB response: Agreed. Change made as above.

L267: To be consistent the ArcticDEM differencing error should be included again here.
CB response: Agreed. Change made as above.

L323: 'most' would be better as 'best'.
CB response: changed as above.

L332: 'physically based depth measurements' should be 'empirically based depth estimates' (or something along those lines). These are not really measurements as such, but rather estimates (here and in the following section).
CB response: changed to "physically based depth estimates"

L349: Is there a missing 'in' in this sentence?
CB response: changed as above.

Referee # 2 comments:

L 13: change "demonstrating" to "which demonstrates"
CB response: changed as above

L 55: I suggest adding a reference to Lampkin et al. 2020 here

(https://www.frontiersin.org/articles/10.3389/feart.2020.00370/full)
CB response: changed as above.

L 57 and 60: change "suggested" to "suggests"
CB response: incorporated into edits from Referee #1

L 122: I suggest adding a reference to Dunmire et al. 2020 here
(https://agupubs.onlinelibrary.wiley.com/doi/10.1029/2020GL087970)
CB response: added above

L 138: Sentence that begins with "Lakes were also checked" is confusing
CB response: edited sentence to read " Time series were also checked for a dip in backscatter prior to the large rise (see 'C' in Figure \ref{fig:timeseriesprocessing}). In the instances where the magnitude of the dip was greater than 25% of the magnitude of the sudden increase, the lake was removed from consideration as a draining lake."

L 161: Section 2.5: I suggest renaming this section "Elevation change from photoclinometry". When I first read the section header I thought that both photoclinometry and DEM differencing would be included in this section. I think

the "and elevation change" is somewhat misleading.
CB response: Changed as above

L 162: Change "This technique: to "photoclinometry:
CB response: Changed as above

L 236: Change "signifying" to "signify"
CB response: Changed to "that signify".

Figure 3: In the caption, please comment that the red outlines signify the lake margins detected from optical imagery.
CB response: changed as above

L 266: Section 3.3: I suggest changing "the ArcticDEM" to "ArcticDEM differencing" in the section heading
CB response: changed as above

L 281: The sentence beginning with " The use of an NDWI_ice – based mask..." seems out of place in this paragraph and I got lost while reading this part.
CB response: Sentence replaced with "Delineating lakes based on optically visible water means that the lake outlines may omit possible subsurface water obscured by an ice lid."

Figure 7: I suggest using a non-diverging color scale when indicating elevation changes between 0 and 12.
CB response: change made as above

Figure 8: It is more appropriate to use a diverging color map that is symmetric (centers on 0). At first glance it appears that the green indicates an increase in elevation everywhere outside the lake but I believe this is not the case?
CB response: changed as above.

L 229: Change "changes in SAR backscatter" to "SAR backscatter changes".
CB response: changed as above.

L 469: Appendix section A4: What do you mean by "Process" for this section header?
CB response: changed to "Photoclinometry point sampling"

Appendix Figure B1: Again, I suggest using a non-divergent color scale to indicate velocity between 0 and 1000 m/yr.
CB response: changed as above.

Appendix Figure C1: Comment that the red boxes surround the lakes in this study.
CB response: changed as above.